# Integrating machine learning with otolith isoscapes: Reconstructing connectivity of a marine fish over four decades

**Kohma Arai**[1]*, **Martin Castonguay**[2], **Vyacheslav Lyubchich**[1], **David H. Secor**[1]

**1** Chesapeake Biological Laboratory, University of Maryland Center for Environmental Science, Solomons, MD, United States of America, **2** Fisheries and Oceans Canada, Institut Maurice-Lamontagne, Mont-Joli, QC, Canada

* karai@umces.edu

## Abstract

Stable isotopes are an important tool to uncover animal migration. Geographic natal assignments often require categorizing the spatial domain through a nominal approach, which can introduce bias given the continuous nature of these tracers. Stable isotopes predicted over a spatial gradient (i.e., isoscapes) allow a probabilistic and continuous assignment of origin across space, although applications to marine organisms remain limited. We present a new framework that integrates nominal and continuous assignment approaches by (1) developing a machine-learning multi-model ensemble classifier using Bayesian model averaging (nominal); and (2) integrating nominal predictions with continuous isoscapes to estimate the probability of origin across the spatial domain (continuous). We applied this integrated framework to predict the geographic origin of the Northwest Atlantic mackerel (*Scomber scombrus*), a migratory pelagic fish comprised of northern and southern components that have distinct spawning sites off Canada (northern contingent) and the US (southern contingent), and seasonally overlap in the US fished regions. The nominal approach based on otolith carbon and oxygen stable isotopes ($\delta^{13}C/\delta^{18}O$) yielded high contingent classification accuracy (84.9%). Contingent assignment of unknown-origin samples revealed prevalent, yet highly varied contingent mixing levels (12.5–83.7%) within the US waters over four decades (1975–2019). Nominal predictions were integrated into mackerel-specific otolith oxygen isoscapes developed independently for Canadian and US waters. The combined approach identified geographic nursery hotspots in known spawning sites, but also detected geographic shifts over multi-decadal time scales. This framework can be applied to other marine species to understand migration and connectivity at a high spatial resolution, relevant to management of unit stocks in fisheries and other conservation assessments.

## Introduction

Migration is ubiquitous across a wide range of taxonomic groups and is among the most spectacular phenomena in nature [1]. Migrations regulate population structure and species

**Data Availability Statement:** All mackerel otolith oxygen and carbon stable isotope measurement

data are available in the Dryad repository: doi:10.
5061/dryad.b8gtht7gr.

**Funding:** Project funding was provided by Fisheries
and Oceans Canada (DFO) under grant number
F5211-190215 to DHS. KA was financially
supported by a fellowship from Japan Student
Services Organization. The funders had no role in
study design, data collection and analysis, decision
to publish, or preparation of the manuscript.

**Competing interests:** The authors have declared
that no competing interests exist.

persistence [2], alter community structure, and facilitate energy flow [3]. Yet, increased anthropogenic activities pose serious threats to highly migratory organisms [4–7]. A better understanding of migration and habitat use of highly migratory species could therefore provide valuable insights for conservation and management, particularly in the face of climate change.

Natural markers (e.g., trace elements, stable isotopes) recorded in biogenic tissues have been employed as a powerful tool to uncover migration and its ecological functions [8–12]. Here, the most common approach to reconstructing migration paths of organisms is by matching the tracer of the organism's tissue to a finite number of potential areas of origin that have been identified *a priori* (i.e., the nominal assignment approach [13]). This method requires that a baseline is established for all known source habitats, and classification methods are applied to assign individuals of unknown origin to their potential source (i.e., the geographic origin is categorical). The categorization of the spatial domain into discrete geographic areas can potentially introduce bias given the continuous nature of tracers, particularly with trace elements and stable isotopes that are distributed throughout the system as a gradient. Natal assignment of the nominal approach can also depend on the classification method of choice [14]. For instance, while technological advances in machine learning-based approaches have shown promise in improving natal assignment performances [15–17], common statistical methods such as discriminant function analysis (DFA) remain prevalent in the literature (but see, e.g., [18–20]).

Recent advances in stable isotope research have allowed probabilistic estimates of natal origin across continuous spatial domains (i.e., the geographic origin is continuous [13]). This less frequent, but rigorous approach seeks to reconstruct migration and connectivity by matching the isotopic signature of the animal tissue to the predicted isotopic composition over a spatial gradient, known as "isoscapes" [21, 22]. In contrast to the nominal assignment approach, "continuous" assignment does not require predefined geographic locations and explicitly incorporates multiple sources of variance-generating processes. Further, isoscapes allow for an in-depth understanding of the biogeochemical processes that give rise to the spatial isotopic gradient, which is frequently ignored in nominal approaches. The continuous assignment approach has been successfully applied to track migration paths of birds, bats, and insects in the terrestrial environment [23]. In comparison, application to marine organisms has been limited to a handful of studies because the isotopic gradient in marine ecosystems varies dynamically over multiple spatiotemporal scales [24–28].

Oxygen and carbon stable isotopes ($\delta^{18}O/\delta^{13}C$ values) in otoliths (or ear stones) have been widely employed as a natural marker to study migration and connectivity of marine fishes (e.g., [10, 29–32]). Otolith aragonite forms near isotopic equilibrium with the oxygen isotopic composition of ambient water, which generally varies globally as a function of salinity [33]. The otolith–water oxygen isotopic fractionation has also been shown to be temperature-dependent, with minor differences across species [34–38]. Otolith $\delta^{13}C$ reflects the carbon isotopic composition of dissolved inorganic carbon (DIC) and the diet source [34, 39, 40], with the proportion of dietary carbon in the otolith controlled by the metabolic rate in a temperature-dependent way [41]. As a result of these features, otolith $\delta^{18}O$ and $\delta^{13}C$ exhibit a predictable spatial gradient, making them suitable for the continuous (isoscape) approach in marine environments.

Here, we present a combined approach that integrates predictions from a nominal assignment approach (machine learning classification) with a continuous assignment approach (isoscapes) to uncover the connectivity and spatial structure of a migratory species in the Northwest Atlantic Ocean over multi-decadal time scales. We first developed a multi-model ensemble classifier as a nominal assignment approach using the Bayesian model averaging

(BMA) framework, where outcomes of multiple machine learning classifiers were aggregated with weights corresponding to their latest prediction performance [42–44]. Then, categorical predictions from the nominal approach were integrated into a species-specific isoscape to estimate the probability of origin across the entire spatial domain using a Bayesian assignment framework (continuous assignment approach). This two-part process emulates the widely applied delta (or hurdle) model approach, wherein the probability of zero observations and non-zero values within the same dataset are modeled separately using two independent models [45, 46]. We demonstrate that the integration of powerful machine learning algorithms with continuous isoscapes allows for an accurate estimation of connectivity and spatial structure of a migratory marine species at a fine spatial-temporal resolution. We further discuss the broader application of our new analytical framework to identify stock structure and habitat use of migratory species in marine ecosystems, and the implications for fisheries stock assessment and management.

## Material and methods

### Study species

The Northwestern population of the Atlantic mackerel (*Scomber scombrus*) is a migratory and pelagic schooling species that support important fisheries in the Northwest Atlantic Ocean. Landings within the US waters have fluctuated greatly, with an intensive foreign fleet fishery occurring in the 1970s reaching historically high landings ($> 400,000$ metric tons); however, the population is currently in a depleted phase experiencing a marked decrease in landings and spawning stock biomass ($< 20,000$ metric tons, [47, 48]). The Northwest Atlantic mackerel population consists of two sub-components: the northern and southern "contingents" that exhibit distinct spawning and nursery regions, as well as seasonal migration patterns that transverse the border between the US and Canada [49, 50]. The two nations separately manage jurisdictional fisheries with Canada assessing the northern contingent as a single stock, and the US treating the two contingents as components of a single transboundary stock [47, 48]. While the estimated median age-at-maturity is about 2 years for both contingents, the spawning stock biomass of the northern contingent is roughly an order of magnitude larger than that of the southern contingent [47, 48, 51, 52]. The dominant northern contingent spawns primarily in the southern Gulf of St. Lawrence in June and July [53–56]. The subordinate southern contingent spawns off southern New England and the Gulf of Maine in April and May [53, 57], although the spawning distribution and nursery habitats of the southern contingent have shifted northeastward over the past four decades [52, 58, 59] (Fig 1). The distribution of the two contingents is mostly separate during the summer spawning season, yet contingent mixing is prevalent throughout late fall, winter, and spring when the northern contingent migrates south and mixes with the southern contingent in US shelf waters where both contingents are subjected to exploitation [50, 60] (Fig 1). Strong latitudinal gradients of SST and water chemistry across Northwest Atlantic shelf waters have recently allowed otolith chemical and stable isotope tracers to accurately discriminate between the two contingents, which confirmed past views on the role of contingent mixing on Atlantic mackerel population dynamics [61–63]. Mapping the spatial structure of the Northwest Atlantic mackerel at fine spatiotemporal scales will offer insight into the assessment and management of this valuable yet depleted two-component population.

### Otolith collection and stable isotope analysis

We used a previously published [61, 63] and unpublished dataset consisting of otolith stable oxygen ($\delta^{18}O$) and carbon ($\delta^{13}C$) isotope values of Northwest Atlantic mackerel. Otoliths were

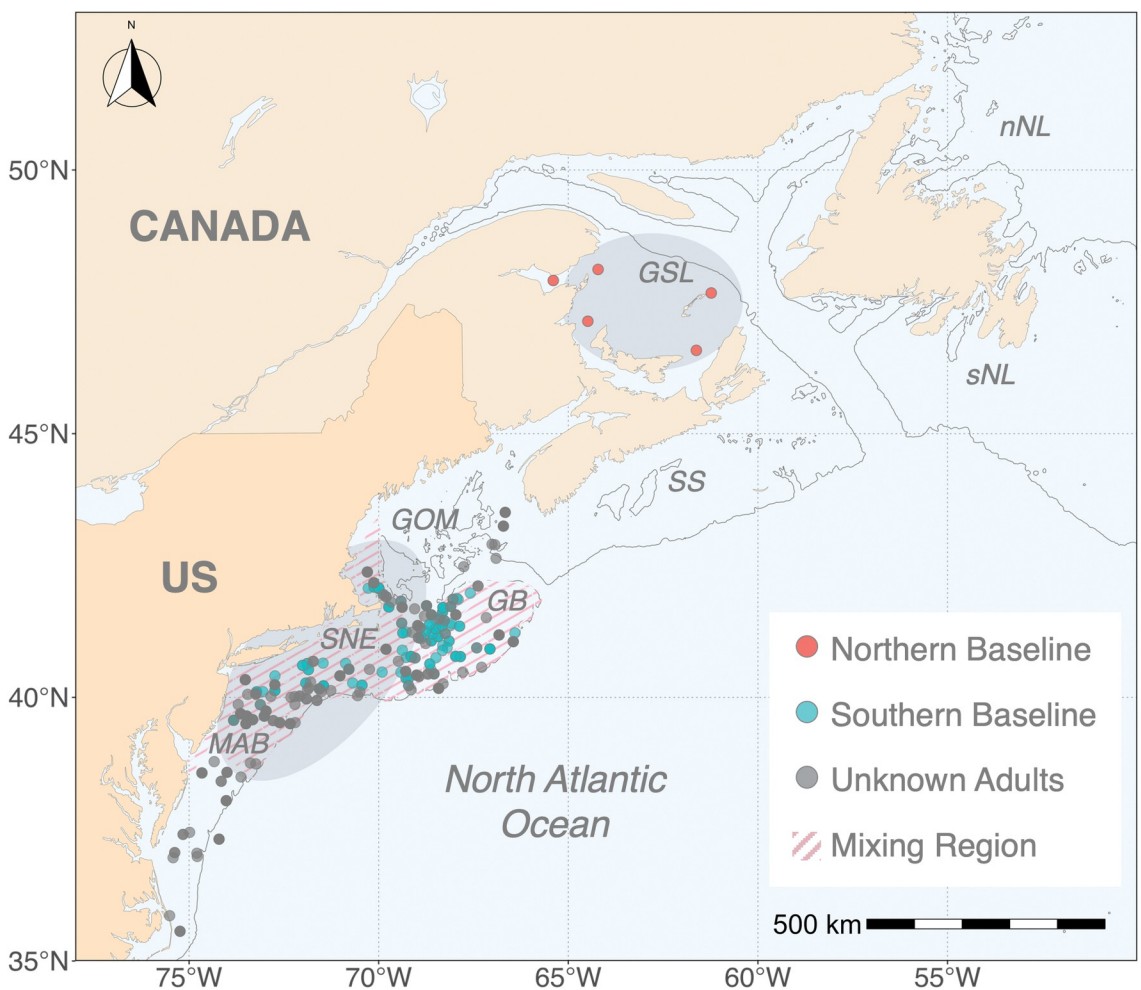

**Fig 1. Map of the western North Atlantic Ocean depicting collection sites of the Northwest Atlantic mackerel.** Shaded ellipses in the southern Gulf of St. Lawrence and US shelf waters indicate principal spawning sites for the northern and southern contingents, respectively. The solid dark line refers to the 200 m isobath. nNL = northern Newfoundland, sNL = southern Newfoundland, GSL = Gulf of St. Lawrence, SS = Scotian Shelf, GOM = Gulf of Maine, GB = Georges Bank, SNE = Southern New England, MAB = Mid-Atlantic Bight. The map was created using the *rworldmap* package [64] in R.

collected in the US winter fishery, the Northeast Fisheries Science Center (NEFSC) fishery-independent spring bottom trawl survey, and the summer Canadian fishery from 1999–2001, 2003, and 2012–2015 ($n$ = 364, [61]) and 2013–2019 ($n$ = 463, [63]) (S1 Table; Fig 1). In addition to the published dataset, we analyzed samples collected in the NEFCS fishery-independent spring bottom trawl survey and the summer Canadian fishery from 1974–1976 and 1978 ($n$ = 135; unpublished; S1 Table; Fig 1).

Detailed methods for otolith processing and stable isotope analysis are available in Redding et al. [61] and Arai et al. [63]. Briefly, we developed a contingent classification baseline using known-origin age-1 juveniles to assign contingent membership to age ≥ 2 unknown-origin adult samples collected within US waters. We assumed that the collection sites of age-1 juvenile samples represented their natal habitats and that the exchange of individuals between the two contingents before reaching adulthood was limited. These assumptions are supported by evidence from size distribution analysis and extensive tagging programs that suggest localized migration patterns of juvenile mackerel [50, 65]. We used a New Wave Research MicroMill

(Fremont, California, USA) to isolate the otolith material within the first annulus representing the early life stage and residence within the natal nursery. Otolith $\delta^{18}O$ and $\delta^{13}C$ values of additional samples included in this study were measured using an automated carbonate preparation device (KIEL-III; Thermo Fisher Scientific, Inc., Bremen, Germany) interfaced with a dual-inlet isotope ratio mass spectrometer (IRMS; Finnigan MAT 252; Thermo Fisher Scientific, Inc., Bremen, Germany) at the University of Arizona's Environmental Isotope Laboratory. All isotope values were reported in delta notation with respect to Vienna-Pee Dee Belemnite (VPDB). No live animals were used in this study and no specific permissions were needed for sampling activities as all otolith samples of Atlantic mackerel (not endangered nor protected) analyzed in this study represented archived material collected from commercial fisheries and government surveys.

## Contingent classification methods

The classification baseline was developed using the age-1 juvenile dataset from a total of 11 year-classes (1973, 1974; 1998–2000; 2011–2016; step A in Fig 2). For some year-classes in which the northern contingent baseline was insufficient in sample size, age-2 northern contingent samples were included to augment the baseline. The inclusion of age-2 fish in the northern contingent baseline is supported by evidence that immigration of the southern contingent into Canadian waters is uncommon [50, 60]. Random oversampling (with replacement) was conducted for all year-classes to balance the sample size within each year-class (total baseline sample size: $n$ = 674). A two-way multivariate analysis of variance (MANOVA) was conducted to test joint differences in otolith $\delta^{18}O$ and $\delta^{13}C$ values between contingents, year-classes, and interaction. Assumptions of MANOVA were verified based on scatterplots of $\delta^{18}O$ and $\delta^{13}C$ on each grouping variable (i.e., contingent and year-class) prior to analysis, and statistical significance was assessed based on Pillai's trace statistic which has been shown to be robust to any violations of assumptions [66, 67]. A post-hoc multivariate pairwise comparison with Bonferroni adjustment was performed to test significant differences in $\delta^{18}O$ and $\delta^{13}C$ between contingents nested within each year-class using the "*mvpaircomp*" function in the "*biotools*" package [68].

We assessed a total of 10 classifiers for contingent membership assignment: four basic statistical methods (Logistic Regression [LR], Mixed-Effect Logistic Regression [MELR], Linear Discriminant Analysis [LDA], and Quadratic Discriminant Analysis [QDA]), and six machine learning methods (Artificial Neural Networks [ANN], Decision Trees [DT], $k$-Nearest Neighbors [$k$NN], Naïve Bayes [NB], Random Forest [RF], and Support Vector Machines [SVM]). For all classifiers, otolith $\delta^{18}O$ and $\delta^{13}C$ were included as predictors to estimate the probability of contingent membership (northern or southern) for each individual. Both predictors were standardized to $z$ scores to facilitate model fitting. "Year-class" was included as an additional variable to account for inter-annual variations in otolith stable isotope values [63]. For the MELR classifier, we allowed all predictors to have separate slopes for each year-class (i.e., random slopes). Hyperparameter tuning of machine learning classifiers was conducted through the grid search method on 10 cross-validated folds to find the optimal hyperparameters (S2 Table). Further, an intercept-only "null" model was included to benchmark the classification performance of all classifiers. Model fitting was performed using the *tidymodels* [69] and *lme4* [70] R packages.

**Multi-model ensemble classification using Bayesian model averaging.** A weighted multi-model ensemble classifier was developed using Bayesian model averaging (BMA). BMA was applied to aggregate predictions from $K$ top-performing classifiers ($k$ = 1, . . ., $K$) based on weights associated with the most recent classification performance [44]. Weights for each

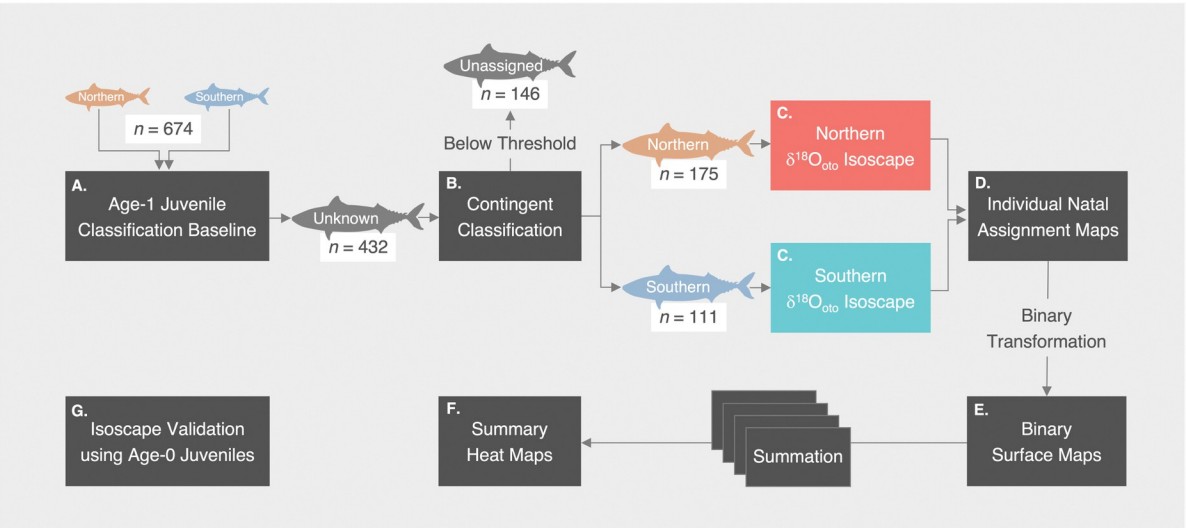

**Fig 2. Schematic of the steps followed to assign contingent membership and to produce geographic natal assignment maps for the Northwest Atlantic mackerel using Bayesian model averaging (BMA) classification and otolith oxygen ($\delta^{18}$O) isoscapes.** (A) Contingent classification baseline development using known-origin age-1 juvenile samples. (B) Binary contingent classification (northern or southern) of age $\geq 2$ unknown adult samples using Bayesian model averaging (BMA) classifier based on otolith stable oxygen ($\delta^{18}$O) and carbon ($\delta^{13}$C) isotope values. Individuals below the classification probability threshold were excluded from further analyses. (C) Binary-classified adults integrated into spatiotemporally resolved contingent-specific otolith $\delta^{18}$O isoscapes generated by coupling sub-surface temperature and salinity (seawater $\delta^{18}$O) data with the chub mackerel fractionation equation. (D) Computing individual probability-of-origin surfaces of binary-classified adults using a Bayesian assignment framework. (E) Binary transformation of individual probability-of-origin surfaces to produce binary surface maps comprising grid cells with a "likely" or "unlikely" location of origin using a defined threshold. (F) Summation of individual binary surface maps for each year-class to produce summary heat maps representing the total number of fish assigned to that location using a defined threshold. (G) Performance validation of the otolith oxygen isoscape using age-0 juvenile samples.

model were proportional to the cross-validated classification accuracy (i.e., the number of correctly classified samples out of all samples) and were normalized so that the weights summed up to 1. Weights were defined as:

$$w(k) = \frac{\text{Accuracy}(k)}{\sum_{k=1}^{K} \text{Accuracy}(k)} \tag{1}$$

where Accuracy($k$) corresponds to the $k$th classifier. Let the predicted values from each classifier be $Pred(k)$ for $k = 1, \ldots, K$. The BMA prediction (i.e., probability of contingent membership) with Accuracy as weights was given by:

$$\text{Pred}_{\text{BMA}} = \sum_{k=1}^{K} w(k)\text{Pred}(k). \tag{2}$$

The performance of all classifiers, including the BMA classifier, was assessed through 10-fold cross-validation using three evaluation metrics: classification accuracy, Area Under the Receiver Operating Characteristic Curve (AUC), and Logarithmic-Loss. Higher classification accuracy and AUC, and lower Logarithmic-Loss imply better classification performance. The classifier with the highest overall performance was selected to assign contingent membership (northern or southern) to age $\geq 2$ unknown adults collected within US waters (step B in Fig 2). We employed a $> 0.7$ classification probability threshold to assign contingent membership, and those below the threshold were unassigned and excluded from geographic natal assignment analysis [63].

**Otolith oxygen isoscape.**    Otolith oxygen isoscapes were generated to make geographic natal assignments of binary-classified adults (step C in Fig 2). Contingent-specific isoscapes

were generated to account for temporal differences in spawning dates and the first growing season: a northern isoscape spanning Newfoundland (NL), Gulf of St. Lawrence (GSL), and Scotian Shelf (SS) was applied to assign individuals classified as the northern contingents, and the southern isoscape spanning the Gulf of Maine (GOM), Georges Bank (GB), Southern New England (SNE), and Mid-Atlantic Bight (MAB) was applied to those assigned the southern contingents. The $\delta^{18}O$ values of otoliths ($\delta^{18}O_{oto}$) can be described by a simple linear fractionation equation with the $\delta^{18}O$ ($\delta^{18}O_{seawater}$) and temperature ($T$) of ambient seawater, with species-specific constant $\gamma$ and $\beta$ parameters:

$$\delta^{18}O_{oto} - \delta^{18}O_{seawater} = \gamma T + \beta. \tag{3}$$

We generated a spatiotemporally resolved mackerel-specific otolith oxygen isoscape by coupling temperature and $\delta^{18}O_{seawater}$ estimates of seawater with an empirically driven congeneric chub mackerel (*Scomber japonicus*) fractionation equation [71]. Isoscapes were generated at a seasonal resolution by averaging over months that represented the period with the highest growth during the first year of life. We averaged temperature and $\delta^{18}O_{seawater}$ for three consecutive months from the peak spawning date for each contingent (southern: May–July; northern: June–August, [53–57]), given that rapid growth occurs up to 90-days post-hatch [72, 73]. To account for inter-annual variations in temperature and $\delta^{18}O_{seawater}$ experienced among individuals of different year-classes, we generated seasonal otolith $\delta^{18}O$ isoscapes for each year that corresponds to all the year classes in which empirical measurements of $\delta^{18}O_{oto}$ values were available (i.e., 1973, 1974; 1998–2000; 2011–2016).

Otolith $\delta^{18}O$ isoscapes were generated using temperature and salinity data at 5 m depth from the EN4 of the Met Office Hadley Centre "EN" series (version EN 4.2.2, [74]), which comprises global temperature and salinity data from 1900 to the present at a monthly/1° spatial grid resolution with 42 depth levels (downloaded February 3, 2022). Data at 5 m depth were used to closely reflect the conditions experienced during the early life stage of Atlantic mackerel during the summer growing season [51]. A North Atlantic Ocean mixing line from LeGrande and Schmidt [33] was used to estimate $\delta^{18}O_{seawater}$ values across the Northwest Atlantic Ocean (assuming linear mixing lines), except for the Gulf of St. Lawrence region where a large freshwater influx from the St. Lawrence River results in distinct salinity–$\delta^{18}O_{seawater}$ relationships from the rest of the Northwest Atlantic Ocean [75]. The following salinity–$\delta^{18}O_{seawater}$ mixing lines were used:

$$\delta^{18}O_{seawater\ (VSMOW)} = 0.55 \times S - 18.98 \text{ (North Atlantic)} \tag{4}$$

$$\delta^{18}O_{seawater\ (VSMOW)} = 0.27 \times S - 10.3 \text{ (Gulf of St. Lawrence)} \tag{5}$$

where $\delta^{18}O_{seawater}$ is the $\delta^{18}O$ [‰ Vienna Standard Mean Ocean Water (VSMOW) scale] of seawater and $S$ is the salinity. Otolith $\delta^{18}O$ values for each point on the 1° × 1° grid were then estimated using a fractionation equation of chub mackerel (*Scomber japonicus*) [71]:

$$\delta^{18}O_{oto\ (VPDB)} = -0.25T + 4.46 + \delta^{18}O_{seawater\ (VSMOW)} \tag{6}$$

where $\delta^{18}O_{oto}$ is the otolith $\delta^{18}O$ [‰ Vienna-Pee Dee Belemnite (VPDB) scale] and $T$ is the temperature (˚C). This resulted in a total of 11 years of mackerel-specific otolith $\delta^{18}O$ isoscapes estimated across known habitats for each contingent representing the highest growth period of Atlantic mackerel (1° grid resolution).

**Geographic natal nursery assignment.** We calculated individual probability-of-origin density surfaces for each binary-classified adult sample (northern or southern) using otolith

$\delta^{18}$O isoscapes (step D in Fig 2). Geographic natal assignments were made using isoscapes specific to their estimated natal region: a northern isoscape for northern contingent samples, and a southern isoscape for southern contingent samples. Probability-of-origin density surfaces were calculated following the approach by Wunder [76], assuming otolith $\delta^{18}$O values to be normally distributed at each grid point and calculating the probability of obtaining the measured isotopic information of the adult sample from normal distributions with an expected mean parameter derived from the isoscapes. The variance model defining the probability distribution of the otolith $\delta^{18}$O value at any given location included (i) analytical error and (ii) within-population variance, which was jointly modeled assuming normality and independence using the following equation [76–78]:

$$\sigma_{combined} = \sqrt{\sigma^2_{analytical} + \sigma^2_{within-pop}} \tag{7}$$

where the $\sigma^2_{analytical}$ was defined as the repeated measurements of NBS-19 standards from the IRMS (0.1‰), and $\sigma^2_{within-pop}$ reflected the variance of otolith $\delta^{18}$O values among age-0 individuals captured at the same known-origin location (0.29‰, S. G. Redding; unpublished data). We then applied Bayes' rule to invert the conditional probability of the measured otolith $\delta^{18}$O values, given the sampling location:

$$P(A_i|B) = \frac{P(B|A_i)P(A_i)}{\sum_i P(B|A_i)P(A_i)} \tag{8}$$

where $P(A_i|B)$ is the posterior probability distribution of a fish originating in geographic location $i$, given the measured otolith $\delta^{18}$O value. $P(B|A_i)$ is the likelihood probability distribution of obtaining the observed otolith $\delta^{18}$O value at geographic location $i$, and $P(A_i)$ is the prior probability distribution of a fish originating in geographic location $i$. The denominator integrates into a constant, and thus the probability density function becomes proportional to the numerator. A uniform prior probability equal to 1 was used for this analysis, such that all locations within the isoscape were equally likely. However, natal origin assignments were constrained within the continental shelf region where most mackerel spawning takes place [51, 53]. The above equation reduces to the following likelihood term:

$$P(A_i|B) \propto L\left(y|\mu_i, \sigma^2_{combined}\right) = \frac{1}{\sqrt{2\pi\sigma^2_{combined}}} e^{\frac{-(y-\mu_i)^2}{2\sigma^2_{combined}}} \tag{9}$$

where $i$ is the geographic location within the Northwest Atlantic, $y$ is the measured otolith $\delta^{18}$O value of the binary-classified adult sample, $\mu_i$ is the predicted otolith $\delta^{18}$O value at location $i$ from the isoscape, and $\sigma^2_{combined}$ is the combined variance defined above. After posterior probabilities were calculated across the region, the value at each cell was divided by the sum of all probabilities so that all cells would sum to 1. Then, the value at each cell was divided by the maximum posterior probability to enable comparisons across individuals.

To assess multi-decadal trends in population-level geographic nursery hotspots, individual probability-of-origin density surfaces were summarized through the binary transformation approach by designating each grid cell as a "likely (1)" or "unlikely (0)" location of origin using a defined threshold (step E in Fig 2, [24, 78]). Grid cells with posterior probabilities in the upper 25% of all grid cells were assigned "likely" and the remaining 75% as "unlikely" (see *Accuracy and precision of the otolith oxygen isoscape* section for details). Individual binary assignment surfaces were summed for each year-class, which produced summary maps with grid cells representing the total number of fish assigned to that location using the defined threshold (step F in Fig 2). Summary maps for each year-class were then divided by the total

number of fish for a given year-class so that values ranged from 0 to 1. For visualization purposes, all maps were interpolated using bilinear interpolation.

**Accuracy and precision of the otolith oxygen isoscape.** Age-0 juveniles collected in the NEFSC fishery-independent bottom trawl survey from September to October 1996, 2001, and 2003 in the MAB, SNE, GB, and GOM provided a unique opportunity to validate the performance of the southern otolith $\delta^{18}$O isoscape (step G in Fig 2; S3 Table; S1 Fig; $n$ = 59, S. G. Redding, unpublished data). We assumed collection sites of age-0 juveniles to closely represent their natal nurseries given the highly localized migration patterns of age-0 juveniles [50, 65]. Isoscapes for 1996, 2001, and 2003 were generated and geographic natal assignments were made for each age-0 juvenile following the steps defined above. Individual probability-of-origin surfaces were transformed into binary surfaces by designating each grid cell as likely (1) or unlikely (0) through a 75th percentile threshold (see previous section, [24, 78]). Individual fish were deemed correctly assigned when grid cells within their collected subregion (e.g., MAB, SNE) included likely grid cells. Accuracy was defined as the proportion of correctly assigned samples to their known collected subregion, and precision (i.e., threshold) was the proportion of grid cells that were assigned as likely among all grid cells. There is often a trade-off between accuracy and precision: defining a stringent threshold (high precision) reduces the number of likely grid cells and thus leads to lower accuracy, and vice-versa [24, 77, 78]. As such, the threshold was determined by assessing the accuracy of the isoscape across a range of precision thresholds (5 to 95%), and the threshold that provided the best accuracy-precision balance was selected.

## Results

### Baseline otolith stable isotope composition

Significant differences of baseline otolith $\delta^{18}$O and $\delta^{13}$C were detected between contingents (Two-way MANOVA, $p_{contingent} < 0.001$; Fig 3), and across year-classes ($p_{year-class} < 0.001$), with a significant interaction effect between contingent and year-class ($p_{contingent:year-class} < 0.001$). Multivariate pairwise comparison between contingents nested within each year-class detected significant differences in all year-classes except for 1974 and 2016 (Fig 3). Baseline otolith $\delta^{18}$O and $\delta^{13}$C values of year-classes 1973 and 1974 showed less distinction between contingents compared to those from year-classes 1998–2000 [61], but exhibited similar patterns to those from the recent year-classes (2011–2016, [63]) in which the northern contingent generally exhibited lower $\delta^{18}$O and higher $\delta^{13}$C values compared to the southern contingent.

### Classification performance

The performance of individual classifiers was grouped into three broad categories based on cross-validated mean classification accuracy: low (accuracy < 0.6: LDA, LR, and NB), moderate (accuracy < 0.76: QDA and DT), and high (accuracy > 0.76: MELR, SVM, ANN, RF, and $k$NN) performance (Fig 4). Note, however, that classification performance was generally consistent across all three metrics. Thus, predictions from high-performing classifiers ($K$ = 5) were aggregated into the multi-model ensemble classifier using BMA. Overall, the BMA classifier outperformed all other individual methods, showing one of the highest classification accuracies (mean: 84.9%), and achieving the highest AUC (mean: 0.93) and lowest logarithmic-loss (mean: 0.38) values.

Using the > 0.7 threshold probability level, the cross-validated mean classification accuracy of the BMA classifier was 66.7, 97.8, and 94.3% for year-classes 1973–1974, 1998–2000, and 2011–2016, respectively (Fig 3; Table 1). The classification performance of the BMA classifier for year-classes 1998–2000 was comparable to previous approaches using Random Forest

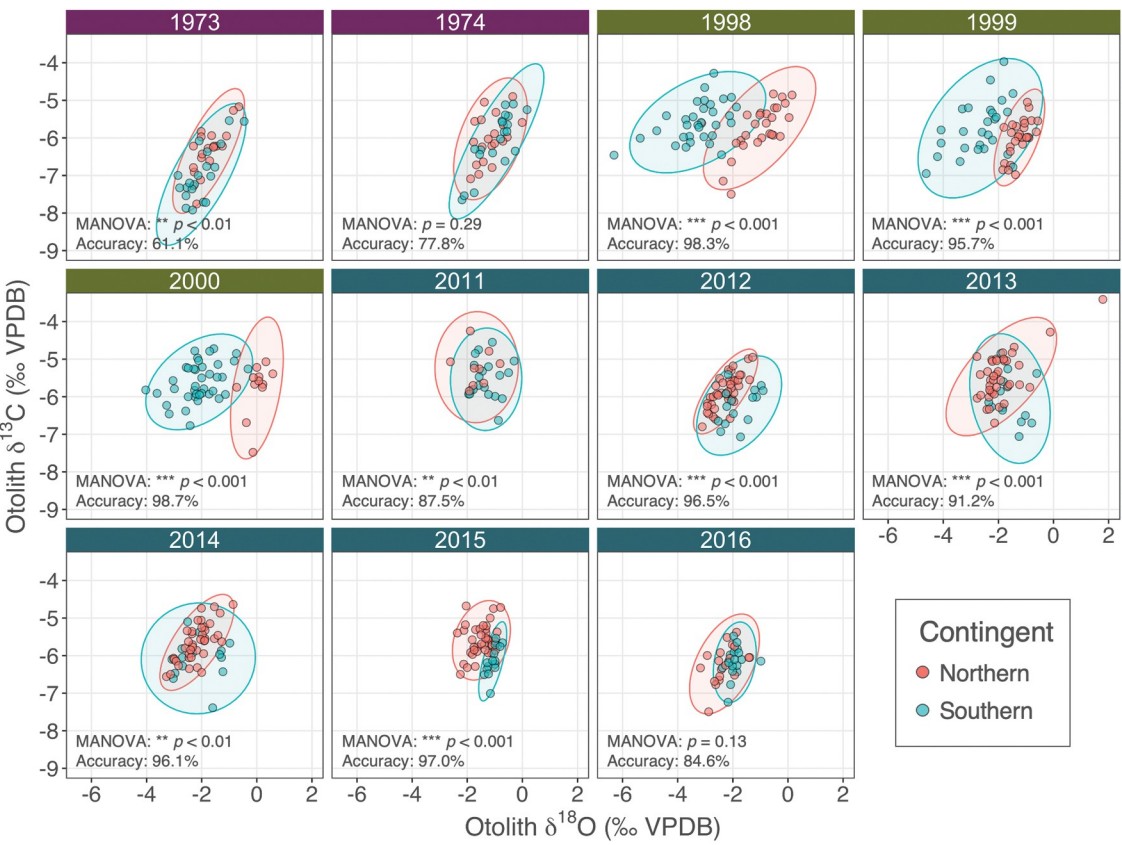

**Fig 3. Northwest Atlantic mackerel baseline otolith oxygen and carbon stable isotope values ($\delta^{18}O/\delta^{13}C$).** Northern and southern contingent baselines are shown for year-classes 1973, 1974 (purple headers), 1998–2000 (green headers), and 2011–2016 (blue headers). 95% confidence ellipses are provided for each contingent. Results from two-way multivariate analysis of variance (MANOVA) pairwise comparison and cross-validated mean classification accuracy with a threshold probability level of 0.7 are shown for each year-class. Data for year-classes 1998–2000 and 2011–2016 are from Redding et al. [61] and Arai et al. [63].

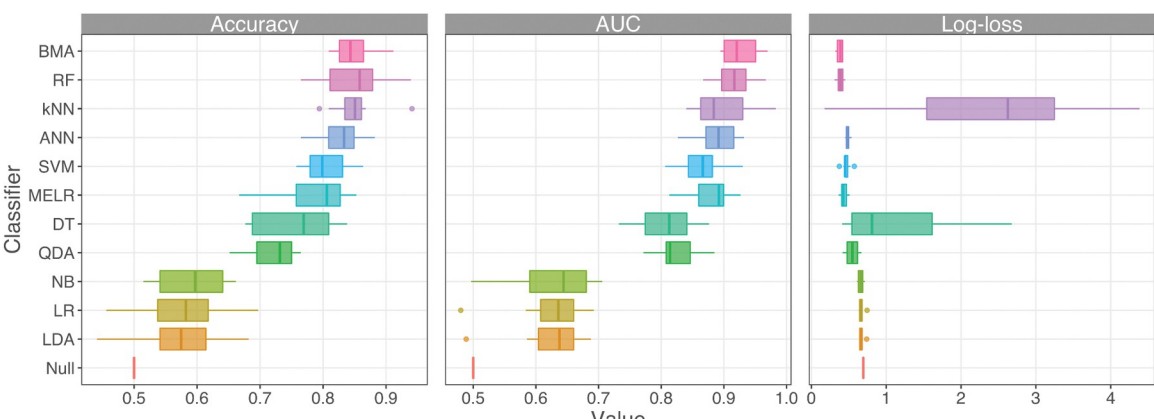

**Fig 4. Classification performance metrics for 12 different classifiers from 10-fold cross-validation.** Performance metrics include classification accuracy (Accuracy, left), Area under the Receiver Operating Characteristic Curve (AUC, middle), and Logarithmic-Loss (Log-loss, right). BMA = Bayesian model averaging classifier, RF = Random Forest, *k*NN = *k*-nearest neighbors, ANN = Artificial Neural Networks, SVM = Support Vector Machines, MELR = Mixed-Effect Logistic Regression, DT = Decision Trees, QDA = Quadratic Discriminant Analysis, NB = Naïve Bayes, LR = Logistic Regression, LDA = Linear Discriminant Analysis, Null = Intercept-only "null" classifier. Note that the scales of the x-axis differ across metrics.

**Table 1. Summary of classification accuracy, and percentage of unassigned baseline and adult samples using the Bayesian model averaging (BMA) classifier.** Cross-validated classification accuracy is shown for probability thresholds 0.7 (outside parenthesis) and 0.5 (inside parenthesis). Classification accuracy and percentage of unassigned baseline and adult fish are also shown for the mixed-effect logistic regression classifier that included otolith $\delta^{18}$O and $\delta^{13}$C as predictors and accounted for year-class effects as a random intercept [63], and the Random Forest classifier that relied only on $\delta^{18}$O values [61].

| Model | 1973–1974 year-class | | | 1998–2000 year-class | | | 2011–2016 year-class | | |
|---|---|---|---|---|---|---|---|---|---|
| | % accuracy | %unassigned (baseline) | %unassigned (adult) | % accuracy | %unassigned (baseline) | %unassigned (adult) | % accuracy | %unassigned (baseline) | %unassigned (adult) |
| Bayesian model averaging (This study) | 66.7 (71.2) | 66.2 | 47.3 | 97.8 (97.4) | 8.2 | 12.7 | 94.3 (81.4) | 34.4 | 39.7 |
| Mixed-effect logistic regression (Arai et al. [63]) | – | – | – | 97.9 (96.9) | 4.2 | 10.9 | 79.7 (69.8) | 47.2 | 46.1 |
| Random Forest (Redding et al. [61]) | – | – | – | 74.5–92.3 | – | – | – | – | – |

(74.5–92.3%, [61]) and mixed-effect logistic regression (97.9%, [63]), although the new approach increased classification accuracy by 14.6 percentage points (p.p.) and reduced the percentage of unassigned baseline samples by 12.8 p.p. for samples from year-classes 2011–2016. Thus, the BMA classifier was further employed to assign contingent membership to age $\geq$ 2 unknown adult samples.

**Adult classification and contingent mixing on the US continental shelf.** Contingent composition estimates from the BMA classifier for year-classes 1998–2000 and 2011–2016 generally agreed with previous estimates from Redding et al. [61] and Arai et al. [63], with only minor changes ($<$ 10%) in contingent composition. BMA classification of age $\geq$ 2 adults indicated a strong dominance of the northern contingent for the 1973 and 1974 year-class (Fig 5). Northern contingent mixing within US shelf waters was prevalent in most year-classes, except for year-classes 2011, 2012, and 2016, yet mixing levels varied greatly over the past four decades. The flexible non-linear classification boundary of the BMA classifier, stemming from the fusion of five independent classifiers, allowed the ensemble classifier to successfully handle the strong temporal variability in the baseline $\delta^{18}$O and $\delta^{13}$C values (Fig 5). A threshold classification probability of $>$ 0.7 resulted in a total of 146 unassigned adult samples (33.8% of total samples). The percentage of unassigned adult samples of year-classes 1998–2000 using the BMA classifier was similar (12.7%) compared to a previous method (10.9%) using a mixed-effect modeling approach (Table 1, [63]). However, for year-classes 2011–2016, the percentage of unassigned adult samples was reduced (39.7%) compared to the previous approach (46.1%).

**Otolith oxygen isoscape.** The predicted mackerel-specific otolith oxygen isoscape exhibited a strong latitudinal trend across the Northwest Atlantic Ocean (Fig 6). Within the US shelf waters, low $\delta^{18}$O$_{oto}$ values occurred in the MAB and SNE, with a northeastward increase toward the GOM and GB. Within Canadian waters, low $\delta^{18}$O$_{oto}$ values were predicted within the SS and southern GSL regions, whereas higher values were observed in the southern and northern NL regions. While the broad spatial pattern in $\delta^{18}$O$_{oto}$ values across the Northwest Atlantic Ocean persisted over four decades, strong temporal variation was observed at regional scales, particularly in the southern GSL, the western GOM, SNE, and MAB regions.

## Geographic natal assignment

The accuracy of the isoscape was assessed using age-0 juvenile samples collected in 1996, 2001, and 2003 in US waters (data shown in S1 Fig). For all year-classes, the accuracy was above 60% for a range of thresholds, although accuracy decreased as the threshold became more restrictive toward 1 (S2 Fig). The 75th percentile threshold provided a balanced accuracy-precision

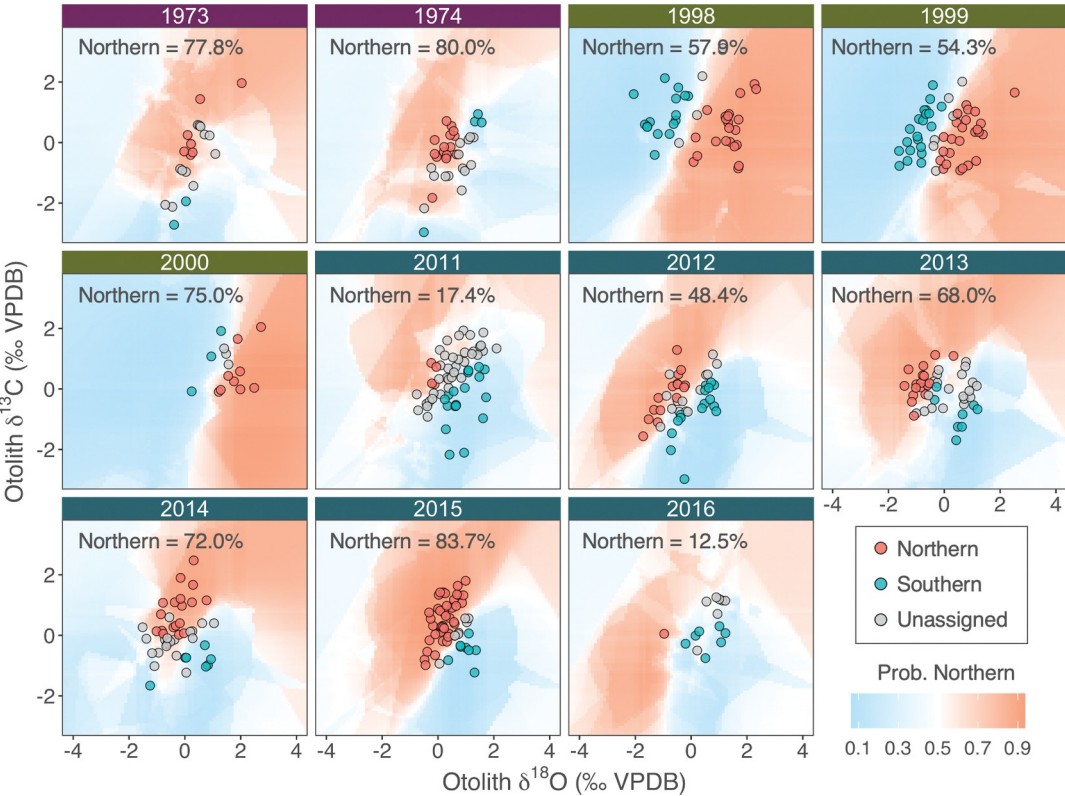

**Fig 5. Northwest Atlantic mackerel otolith oxygen and carbon stable isotope values ($\delta^{18}$O/$\delta^{13}$C) for age $\geq$ 2 unknown adults collected in US waters.** Contingent membership assignment results using the Bayesian model averaging (BMA) classifier are shown for year-classes 1973, 1974 (purple headers), 1998–2000 (green headers), and 2011–2016 (blue headers). Background colors depict the probability of contingent membership (northern or southern). Individuals below the 0.7 classification probability threshold were unassigned. The percentage of the northern contingent of assigned fish (probability > 0.7) for each year-class is shown in the top left corner of each panel. Isotope values are shown as $z$ scores with mean 0 and SD 1.

trade-off that maintained a high percentage of correct assignment (i.e., accuracy) while also constraining the area of geographic assignment (i.e., precision). Thus, the 75th percentile threshold was further employed for binary transformation (i.e., grid cells with posterior probabilities in the upper 25% of the total grid cells were assigned as likely locations). The majority of age-0 juveniles– 70% (1996), 87.5% (2001), and 82.6% (2003)–were correctly assigned to their collected subregions with the selected threshold.

Geographic natal nursery assignments were compiled for binary-classified adult samples using the mackerel-specific otolith oxygen isoscape, which resulted in a total of 111 and 175 individual probability-of-origin surfaces for the southern and northern contingents, respectively (Fig 7). Individual probability-of-origin surfaces were binary-transformed and subsequently summed for each year-class to identify hotspots in nursery habitat use and geographical shifts across multi-decadal scales (Fig 8). Nursery hotspots centered around the southern GSL and SS in most but not all year-classes for the northern contingent (Fig 8A). In year-classes 1998–2000, nurseries were predicted outside the GSL, concentrating around the northeastern and southwestern shelves of NL. For those assigned the southern contingent, natal nurseries generally concentrated in the MAB and SNE across all year-classes, although nursery hotspots expanded northeastward to the western GOM after 2000 (Fig 8B).

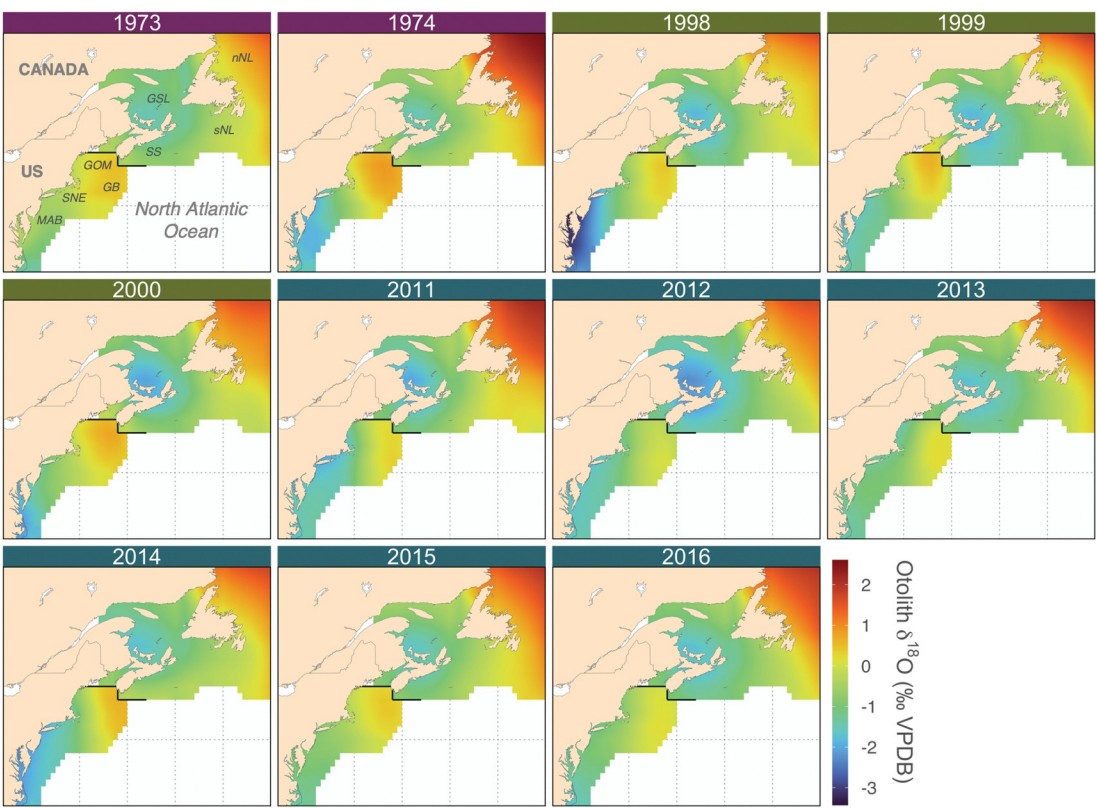

**Fig 6. Mackerel-specific otolith oxygen isoscapes for 11 years.** Isoscapes are depicted for year-classes 1973, 1974 (purple headers), 1998–2000 (green headers), and 2011–2016 (blue headers). Note that the northern and southern isoscapes were generated separately by averaging over months of highest growth during the first year of life of Atlantic mackerel (southern: May–July; northern: June–August) and delineated with the solid black line. nNL = northern Newfoundland, sNL = southern Newfoundland, GSL = Gulf of St. Lawrence, SS = Scotian Shelf, GOM = Gulf of Maine, GB = Georges Bank, SNE = Southern New England, MAB = Mid-Atlantic Bight. The map was created using the *rworldmap* package [64] in R.

## Discussion

By integrating modern machine learning algorithms with continuous otolith $\delta^{18}$O isoscapes, we accurately predicted the geographic origin of a migratory marine species at fine spatiotemporal scales across the Northwest Atlantic Ocean over multiple decades. This new analytical framework should be applicable to a range of migratory species in the marine environment to inform future conservation and management actions.

### Integrating machine learning with isoscapes

Natal assignment of the nominal approach can vary according to the classification method of choice, and often involves selecting a single "best" model, which ignores model uncertainty. The BMA-based multi-model ensemble classifier outperformed all individual classifiers (Fig 4) and substantially improved classification performance compared to previous approaches on the same dataset [61, 63] (Table 1). The BMA approach not only retains all model uncertainty but is also robust to model misspecification as the portfolio of models bet-hedges against misclassification [42, 79]. Further, machine learning classifiers outperformed conventional statistical approaches (e.g., LDA, LR; Fig 4), successfully handled non-linear patterns in the dataset, and were particularly robust to strong temporal variations, which is common in natural tracer-type datasets [9, 80, 81].

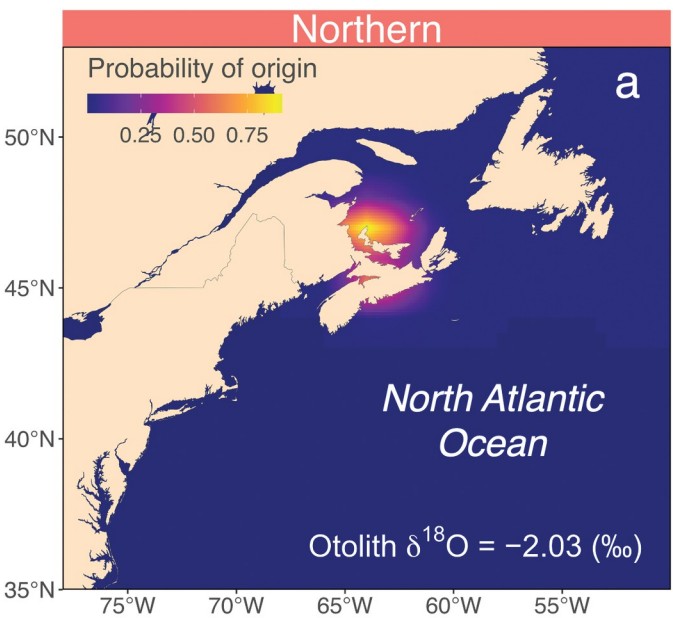

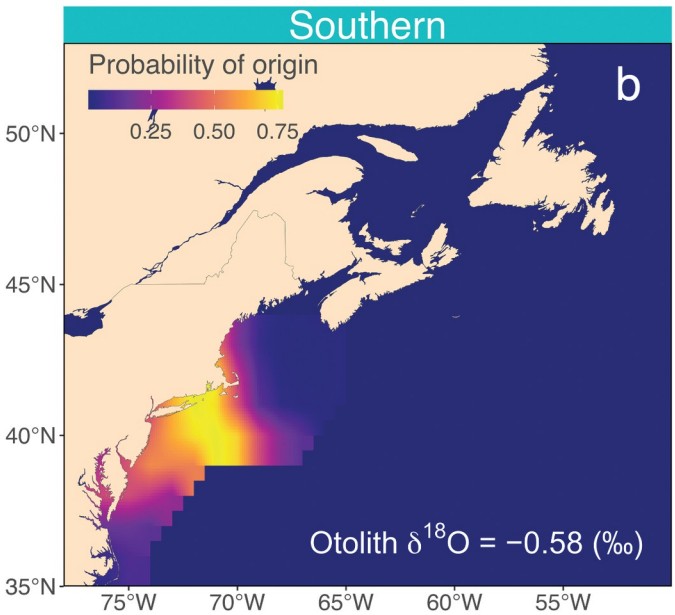

**Fig 7. Example individual probability-of-origin surfaces of Northwest Atlantic mackerel.** Plots include fish assigned the northern ($\delta^{18}O_{oto} = -2.03‰$) and southern ($\delta^{18}O_{oto} = -0.58‰$) contingents from the 2015 year-class. The map was created using the *rworldmap* package [64] in R.

The integration of BMA-based predictions with spatiotemporally resolved isoscapes allowed accurate geographic natal assignment of a migratory marine species at a fine spatial resolution (Fig 7). Previous studies have applied the nominal and continuous approaches independently to address the question of interest [24, 77, 82]. However, instead of applying the two approaches independently, this study leveraged the nominal approach to inform the continuous approach. The nominal approach required predefined geographic locations but was highly

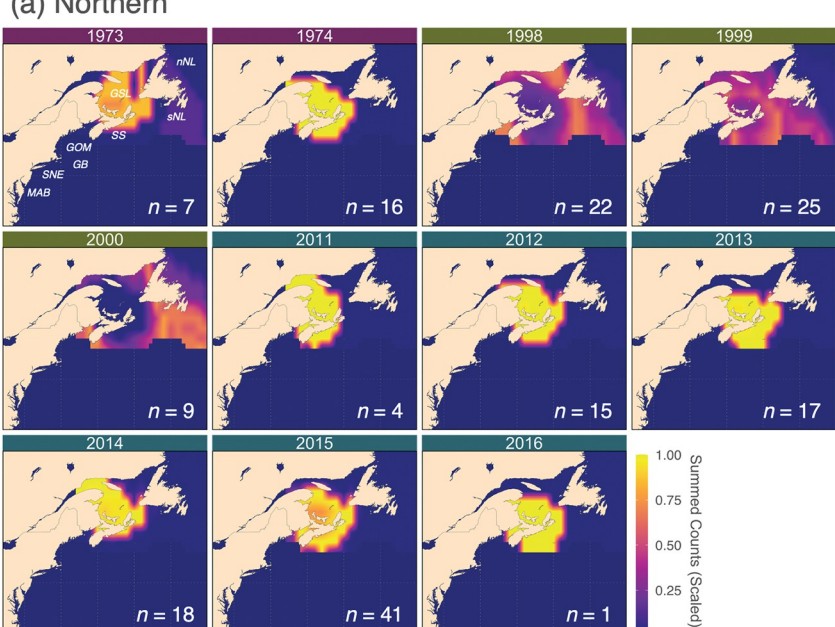

**Fig 8.** Summary heat maps in nursery habitat use for northern (a) and southern (b) contingents. Heat maps are depicted for year-classes 1973, 1974 (purple headers), 1998–2000 (green headers), and 2011–2016 (blue headers). Heat maps were computed by summing individual binary assignment surfaces for each year-class and scaled by the sample size such that all values range from 0 to 1. A 75th percentile threshold was employed for binary surface transformation prior to summation. The sample size included for each heat map for each year-class is also shown. nNL = northern Newfoundland, sNL = southern Newfoundland, GSL = Gulf of St. Lawrence, SS = Scotian Shelf, GOM = Gulf of Maine, GB = Georges Bank, SNE = Southern New England, MAB = Mid-Atlantic Bight. The map was created using the *rworldmap* package [64] in R.

accurate and enabled the inclusion of an informative otolith $\delta^{13}C$ predictor that could be challenging to model across space. The nominal approach further constrained geographic origin predictions of the continuous approach that modeled the otolith tracer as a continuum and explicitly propagated multiple variance-generating processes. The resulting combined framework provided fine-scale geographic origin estimates on predefined locations, which would be particularly informative for management and conservation applications that often require defined boundaries. Compiling individual probability surface maps into summary heatmaps further enabled the detection of population-level geographic nursery habitat shifts over multi-decadal time scales (Fig 8)–a key finding that would have been otherwise overlooked by relying upon the nominal approach alone.

## Spatial structure of Northwest Atlantic mackerel over four decades

Contingent assignment using otolith $\delta^{18}O$ and $\delta^{13}C$ values presented here, and assignments by Redding et al. [61] and Arai et al. [63] revealed prevalent contingent mixing within US waters over four decades (1975–2019), supporting traditional views on the spatial structure of the Northwest Atlantic mackerel [50] (Fig 5). It is important to exercise caution when interpreting contingent mixing levels, as some year-classes (e.g., 1974, 2016) exhibited weak stable isotopic distinction (Fig 3). Nonetheless, the aggregated year-class baseline compensates for temporal variation and enables year-classes with less distinct isotopic signatures to draw information from those that are more distinct, bringing them closer to the global mean. Furthermore, a conservative 0.7 classification threshold was applied to reduce misclassification of unknown adult samples.

Temporal variation in mixing levels could reflect changes in the recruitment strength and migration patterns by the dominant northern contingent [63, 83]. Namely, high mixing levels for year-classes 1973, 1974, and 2015 reflected strong northern contingent cohorts, and low mixing in the 2016 year-class corresponded to historically low recruitment levels [84]. Of particular interest was the presence of the northern contingent during the 1970s –periods when an intensive foreign fleet fishery occurred within US waters. While small sample sizes and high unassignment rates preclude strong inferences for this period, these results could imply that the northern contingent could have provided an important seasonal subsidy to support the large historical fishery.

Nursery hotspots were detected by compiling individual binary surface maps (Fig 8). Northern contingent nursery hotspots generally occurred in the known and monitored spawning site in the southern GSL region [53, 54, 56] (Fig 8A). Interestingly, in 1998–2000, hotspots were identified in the northeastern and southwestern shelves of NL, where spawning activity is small and sporadic [85]. This apparent "spill-over" could be a result of a density-dependent range expansion associated with the exceptionally strong 1999 year-class [86, 87]. However, there is little evidence of increased egg production and spawning stock biomass in the southern GSL in the 2000s following the arrival of the exceptional 1999 year-class [55]. This, combined with our findings suggests that the 1999 year-class may have utilized nurseries outside the GSL, perhaps off southern NL. Density-dependent range expansion has been well-documented for the Northeastern population of the Atlantic mackerel [88, 89], and is further supported by the increase in landings in NL during years of strong recruitment [48]. For the southern contingent, the expansion of nursery hotspots into the western GOM over the recent decade corresponds remarkably well to reported northeastward shifts in adult, egg, and larval habitat distributions, primarily driven by changes in temperature [52, 58, 59, 90] (Fig 8B). As these distribution shifts and complex spatial structure pose challenges for fisheries management, it is vital to further incorporate predictions from habitat suitability models [54, 59] and

egg survey data [48, 52] into our continuous framework to provide fine spatial information on this depleted population. Furthermore, nursery hotspot predictions from year-classes with small sample sizes (e.g., 1973, 2000, 2016) are prone to bias and may not be indicative of the contingent-level nursery habitat and should be considered with caution.

## Spatial patterns in mackerel otolith oxygen isoscapes

Predicted mackerel $\delta^{18}O_{oto}$ values were distributed across the Northwest Atlantic as a gradient, highlighting the suitability of the continuous approach to stable isotope applications (Fig 6). The latitudinal $\delta^{18}O_{oto}$ gradient was consistent with the differences in natal $\delta^{18}O_{oto}$ values between juvenile northern and southern contingent baselines (Fig 3). The spatial pattern of $\delta^{18}O_{oto}$ is largely governed by physical processes including the Gulf Stream, Labrador Current, and the St. Lawrence River, which drive changes in the $\delta^{18}O$ of seawater ($\delta^{18}O_{seawater}$) and temperature across the Northwest Atlantic [91, 92]. In the southern GSL, $^{18}O$-depleted freshwater from the St. Lawrence River combined with warm temperatures leads to a localized pool of low $\delta^{18}O_{oto}$ [75, 93]. The latitudinal $\delta^{18}O_{oto}$ gradient within the US shelf waters is likely governed by the temperature gradient attributed to the Gulf Stream rather than $\delta^{18}O_{seawater}$, resulting in a northeastward increase in $\delta^{18}O_{oto}$ toward the GOM and GB. This spatial pattern corresponds well with sub-regional differences in measured $\delta^{18}O_{oto}$ values of age-0 juveniles collected across the US shelf waters (S1b Fig).

　　Strong inter-decadal variation in both measured juvenile $\delta^{18}O_{oto}$ (Fig 3) and isoscapes (Fig 6) could be related to changes in large-scale oceanographic features, such as the Atlantic meridional overturning circulation (AMOC, [94]). The weakening of AMOC during recent decades has been associated with the northward shift of the Gulf stream position combined with the poleward retreat of the Labrador Current, leading to warmer US shelf waters and consequently lower $\delta^{18}O_{oto}$, particularly in the western Gulf of Maine region [91, 95, 96]. There was no clear inter-decadal pattern in the measured juvenile baseline for otolith $\delta^{13}C$ values (Fig 3); however, the decline in atmospheric $\delta^{13}C$ values has been linked to a significant declining trend in baseline otolith $\delta^{13}C$ in Atlantic bluefin tuna (i.e., Suess effect) [81]. These strong temporal patterns in otolith stable isotopes further highlight the importance of accounting for such variation in the baseline for accurate natal assignment.

## Broader applications and implications to management

The combined approach proposed here requires a reference baseline and a species-specific otolith $\delta^{18}O$ isoscape generated from temperature and salinity data through a species-specific fractionation equation. In particular, $\delta^{18}O$ of ambient seawater ($\delta^{18}O_{seawater}$) was determined from salinity data which could be problematic for species that inhabit regions that are heavily influenced by multiple sources of freshwater inputs, in which the salinity–$\delta^{18}O_{seawater}$ relationship may be highly dynamic [97]. In this case, direct $\delta^{18}O_{seawater}$ measurements should be preferred for otolith $\delta^{18}O$ isoscape development, although the spatiotemporal resolution could be reduced [98]. Furthermore, because otolith $\delta^{13}C$ is influenced by an unknown combination of ambient DIC, food source, and metabolic rates [34, 39, 40], we were unable to establish a spatiotemporally accurate otolith $\delta^{13}C$ isoscape. Instead, we employed otolith $\delta^{13}C$ data as a predictor in the nominal approach, which was then used to inform the continuous approach. As global-scale estimations of these values have become more readily available [99–103], a multi-isotope isoscape model could be developed and applied to provide a more accurate geographic origin estimate in marine species [24, 26].

　　Overall, we provide a promising framework that translates the biogenic stable isotope information into a probabilistic prediction of geographic origin in a marine environment. The

application of the continuous assignment approach to marine species has been limited given the lack of spatial information on geochemical tracers in marine ecosystems (but see, e.g., [28]). The increased accessibility of modern machine learning algorithms [69], combined with biomineral $\delta^{18}O$ isoscapes predicted from readily available data (i.e., temperature and salinity), makes this approach widely applicable to any marine species, but particularly those that form carbonate biominerals (e.g., otoliths). As these hard parts are commonly retained and stored in abundance for fisheries stock assessment purposes, they can be used to infer population-level migration and connectivity of important fishery resources [104, 105]. Archived hard parts could be particularly useful in reconstructing fine-scale migration patterns across decadal time scales to understand the effects of climate change [106, 107]. Understanding migration and connectivity at fine spatial-temporal scales will not only allow fisheries management units to better reflect the spatial structure of biological populations [49, 108–111], but also expand our knowledge on the role of migration in population structuring and species persistence in the face of climate change.

## Supporting information

**S1 Fig.** Collection sites (a) and measured otolith oxygen stable isotope values (b) of age-0 juvenile Northwest Atlantic mackerel collected in 1996, 1999, and 2003. GB = Georges Bank, GOM = Gulf of Maine, SNE = Southern New England, MAB = Mid-Atlantic Bight. The map was created using the *rworldmap* package [64] in R.
(TIF)

**S2 Fig. Geographic natal origin assignment accuracy computed from known-origin age-0 juvenile Northwest Atlantic mackerel collected in US shelf waters in 1996, 2001, and 2003 (*n* = 59).** Accuracy was computed over a range of thresholds (i.e., precision) from 0.05 to 0.95. The 75th percentile threshold (dotted vertical line) provided a balanced accuracy-precision trade-off and was selected for binary transformation, where grid cells with posterior probabilities in the upper 25% of all grid cells were assigned "likely" and the remaining 75% as "unlikely" location of origin.
(TIF)

**S1 Table. Northwest Atlantic mackerel otolith sample size by year-class, sampling period, collection site, and data type.** Data for year-classes 1998–2000 and 2011 are from Redding et al. [61], and data for year-classes 2012–2016 are from Arai et al. [63].
(DOCX)

**S2 Table. Hyperparameters selected for six machine learning classifiers using the grid search method on 10 cross-validation folds.**
(DOCX)

**S3 Table. Age-0 juvenile Northwest Atlantic mackerel otolith sample size by year-class, sampling period, and collection site.**
(DOCX)

## Acknowledgments

We thank Sandy Sutherland of the National Marine Fisheries Service, Northeast Fisheries Science Center (NMFS-NEFSC), and Mélanie Boudreau of Fisheries and Oceans Canada (DFO) for procuring archived Atlantic mackerel otolith samples. Thank you to David Dettman of the Environmental Isotope Laboratory, University of Arizona for facilitating the stable isotope analysis. Thanks to Lisa Kerr and Zachary Whitener of the Gulf of Maine Research Institute,

and Lisa Greer of the Geology Department, Washington and Lee University for generously providing access to the micromill. We thank Gray Redding of the National Fish and Wildlife Foundation for kindly providing age-0 juvenile data. We also wish to thank David Nelson of the University of Maryland Center of Environmental Science for an initial review that significantly improved the manuscript, as well as two anonymous reviewers for their helpful comments.

## Author Contributions

**Conceptualization:** Kohma Arai, Martin Castonguay, David H. Secor.

**Data curation:** Kohma Arai.

**Formal analysis:** Kohma Arai.

**Funding acquisition:** David H. Secor.

**Investigation:** Kohma Arai.

**Methodology:** Kohma Arai, Martin Castonguay, Vyacheslav Lyubchich, David H. Secor.

**Project administration:** Martin Castonguay, David H. Secor.

**Resources:** Kohma Arai, Martin Castonguay.

**Software:** Kohma Arai.

**Supervision:** Martin Castonguay, David H. Secor.

**Validation:** Martin Castonguay, Vyacheslav Lyubchich, David H. Secor.

**Visualization:** Kohma Arai.

**Writing – original draft:** Kohma Arai.

**Writing – review & editing:** Martin Castonguay, Vyacheslav Lyubchich, David H. Secor.

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
