## [Decision Letter · Decision Letter 0]

7 Mar 2023

PONE-D-23-01598

Integrating machine learning with otolith isoscapes: reconstructing connectivity of a marine fish over four decades

PLOS ONE

Dear Dr. Arai,

Thank you for submitting your manuscript to PLOS ONE. After careful consideration, we feel that it has merit but does not fully meet PLOS ONE’s publication criteria as it currently stands. Therefore, we invite you to submit a revised version of the manuscript that addresses the points raised during the review process.

The reviewers feel that your manuscript is a valuable contribution that deserves to be published. However, they have raised some aspects of the manuscript that should be addressed and resolved before the manuscript can be considered for publication.

We look forward to receiving your revised manuscript.

Kind regards,

Antonio Medina Guerrero, Ph.D.

Academic Editor

PLOS ONE

Journal Requirements:

“KA was financially supported by a fellowship from Japan Student Services Organization.”

“Project funding was provided by Fisheries and Oceans Canada (DFO) under grant number F5211-190215 to DHS. The funders had no role in study design, data collection and analysis, decision to publish, or preparation of the manuscript.”

5. We note that Figures 1, 5, 6, 7 and 8 in your submission contain map images which may be copyrighted. All PLOS content is published under the Creative Commons Attribution License (CC BY 4.0), which means that the manuscript, images, and Supporting Information files will be freely available online, and any third party is permitted to access, download, copy, distribute, and use these materials in any way, even commercially, with proper attribution. For these reasons, we cannot publish previously copyrighted maps or satellite images created using proprietary data, such as Google software (Google Maps, Street View, and Earth). For more information, see our copyright guidelines: http://journals.plos.org/plosone/s/licenses-and-copyright.

 a. You may seek permission from the original copyright holder of Figures 1, 5, 6, 7 and 8 to publish the content specifically under the CC BY 4.0 license. 

Reviewers' comments:

Reviewer's Responses to Questions

**Comments to the Author**

1. Is the manuscript technically sound, and do the data support the conclusions?

Reviewer #1: Yes

Reviewer #2: Partly

2. Has the statistical analysis been performed appropriately and rigorously? 

Reviewer #1: Yes

Reviewer #2: Yes

3. Have the authors made all data underlying the findings in their manuscript fully available?

Reviewer #1: Yes

Reviewer #2: Yes

4. Is the manuscript presented in an intelligible fashion and written in standard English?

Reviewer #1: Yes

Reviewer #2: Yes

5. Review Comments to the Author

Reviewer #1: This study proposed and described a novel framework to delineate the geographic origin and nursery hotspots over a spatiotemporal scale for the Northwest Atlantic mackerel by means of otolith stable isotope concentrations, which could be a suitable and interesting approach for other marine species as well.

This paper presents the results of original research with an important contribution to the field of stock management informed otolith microchemistry presented as a high-quality manuscript which significant importance for the broad readership of PLOS ONE journal. The manuscript contains all the relevant information on the field, and is easy to follow, the motivation behind the work is clearly described, with experiments, statistics, and other analyses are performed to a high technical standard and are described in sufficient detail, with straight and sound conclusions that are supported by the data. The manuscript is easy to follow and is presented in presented in an intelligible fashion and is written in standard English. Authors do state that meet all applicable standards for the ethics of experimentation and research integrity, and that all data are fully available without restriction.

However, there are some minor issues that could be improved in my point of view before its publication as a high impact article. I have provided general comments and suggestions for each section of the manuscript, followed by specific comments to the authors that should be quite straightforward to address, I hope authors find them useful. If authors can address these minor issues, I do believe that this manuscript will be of great value as a publication for PLOS ONE journal. Finally, my most sincere congratulations for this nice piece of work, this is good science that needs to be shared!

INTRODUCTION

The introduction provides the reader an adequate background and clearly described the motivation behind this work. However, I do fill that too much weight is given (lines 86-99) on how the analysis has been done, which is perfectly described in M&M sections, and I do miss few more background on otolith stable isotopes composition, e.g. how oxygen composition of otolith aragonite responds to variations in the environment and why it is useful. If you could balance the text between both topics it would be very nice.

Specific comments to the authors:

Line 53: I would suggest replacing “internal tags” for something like “natural markers” as internal tags usually refers to fish tagging devices and can be confusing.

Line 54: I would suggest starting the second sentence with an “Here” or sth similar to emphasize that you are still talking about chemical markers in biogenic tissues, as if not, it could be again misunderstood as the most common approach for tracking highly migratory fish is by archival or pop up tagging. Something like “Here, the most common approach consists of reconstructing…”

Line 63: You can cite Jones et al. 2017 here to reinforce this statement.

Jones, C. M., Palmer, M., & Schaffler, J. J. (2017). Beyond Z ar: the use and abuse of classification statistics for otolith chemistry. Journal of Fish Biology, 90(2), 492-504.

Lines 63-66: This is true because most older papers used traditional approaches, but I would also note that there is a trend towards change e.g.

Wright, P. J., Régnier, T., Gibb, F. M., Augley, J., & Devalla, S. (2018). Assessing the role of ontogenetic movement in maintaining population structure in fish using otolith microchemistry. Ecology and Evolution, 8(16), 7907-7920.

Zhang, C., Ye, Z., Li, Z., Wan, R., Ren, Y., and Dou, S. (2016). Population structure ofJapanese Spanish mackerel Scomberomorus niphonius in the Bohai Sea, the Yellow Sea and the East China Sea: evidence from random forests based on otolith features. Fisheries Science 82, 251–256. doi:10.1007/S12562-016-0968-X

Artetxe-Arrate, I., Fraile, I., Crook, D. A., Zudaire, I., Arrizabalaga, H., Greig, A., & Murua, H. (2019). Otolith microchemistry: a useful tool for investigating stock structure of yellowfin tuna (Thunnus albacares) in the Indian Ocean. Marine and Freshwater Research, 70(12), 1708-1721.

Lines 72-73: Not sure if completely agree, don’t always explicitly incorporate all sources of variation into account. E.g. individual physiology for stable isotope incorporation

Line 80: Perhaps “Combined” instead of “Hybrid”?

Line 82: I think that is not correct to use the term “highly migratory marine species here”. Highly migratory species are legally defined as those listed in Annex 1 of UNCLOS (https://www.un.org/depts/los/convention_agreements/texts/unclos/annex1.htm) and Atlantic mackerel is not part of this list. Please adapt this sentence here and elsewhere in the manuscript.

MATERIAL AND METHODS

The section provides enough detail on the technique, equations and statistical approaches used so that other researchers can reproduce them. I just miss a little bit more information regarding the Study species, such as how it is managed (e.g. Atlantic mackerel managed separately in US and Canada?) and spawning conditions that feel important to known, such as when are they capable of spawning, age or length, so that then it is better understood why you separate age-1 as baseline and age>2 for assignment purposes. I also would recommend to include a summary Table with the individuals that have been analysed, giving information numbers in the, 201 baseline, which are for assignment, how much fish belong to each year, how much fish were collected from each fishery etc and also present the age-0 juvenile samples used for testing the accuracy of the isoscape.

Specific comments to the authors:

Line 102: See comment above about “highly migratory”, perhaps you can find another way to explain this, or go a bit further and explain here the migrations that Atlantic mackerel undergoes (e.g. from spawning grounds in XX towards feeding grounds in XX).

Line 109: What does “dominant” imply here? Higher production, biomass…?

Line 118: I would suggest removing “(or ear stone)” from here, and if you prefer, to move this to the Introduction.

Lines 137 and 139: I cannot see what you are referring to in Fig.1. Perhaps a table could help.

Line 142: Why do you say that age-1 are known origin? Please can you add more information that support this statement, e.g. they do not migrate until age X, or they have shown to have limited movements outside nursery areas….

Lines 158-159: Can you explain more on the implications of including some age-2 northern contingents in the baseline? Where they used again for assignments or not? Does Atlantic mackerel show some spawning site fidelity?

Lines 161: Was Multivariate normality, linearity and multicolinearity and homoscedasticity of otolith �18O and �13C data tested prior MANOVA analyses?

Line 245: If possible, could you please add which depth range are you considering here as “subsurface” (e.g 0-5 m depth)..

RESULTS

Overall results are well presented and support the data, and are reported in a concise, straightforward manner, using tables or figures when appropriate. Figure captions are very well described so that they are easy to interpret.

Specific comments to the authors:

Line 341-343: Are there significant differences between contingents in every year? I think it will be nice an interesting to analyse this option too, as there are years (e.g. 1973, 1974, 2011, 2016…) were there it seems not to be differentiation, and this is also important for the discussion.

Line 343-344: Less distinction or no distinction?

Line 347: From the Figure 3 you cannot appreciate the fact that northern contingents from 1973 1974, 2011,2013 and 2014 for instance had lower oxygen values.

Lines 357: What is the threshold for low moderate and high? Not sure if needed but think it will improve readers experience.

Line 395-396: From figure 5 it seems that northern contingents were not prevalent in 2011, 2012 and 2016, please clarify this.

Lines 400-403: This paragraph is a little bit confusing for me, because I do not have very clear were this numbers come from.

Figure 6: I think it could facilitate to the reader if you can add the locations you mentioned in the text in lines 415-425. This is just a suggestion, but as you have 4 x3 plots and a blank space, you can use this black space to draw the same map without predictions but with location names, so then it is easier to follow the rest of the maps. If not, one can always return to Figure 1.

DISCUSSION

Authors place into context observed results and highlight the most important and discuss the implications of this study into a broader context, which is very interesting and nice. It provides a good overview and closure of the manuscript, with conclusions aligning with the aim of the proposed research. However, I do miss some more further discussion on the potential limitations of the approach and some mention to he Suess effect into otolith C13 data.

Specific comments to the authors:

Line 495: I suggest replacing “hybrid” by “combined”

Line 529: Perhaps “primarily driven by changes in temperature”

Line 559: I suggest replacing “tissue” here by “biogenic”, “hard-calcareous structures” or others here.

Line 562: But see Martino et al. 2022

Martino JC, Trueman CN, Mazumder D, Crawford J, Doubleday ZA. The universal imprint

of oxygen isotopes can track the origins of seafood. Fish Fish. 2022. doi:10.1111/faf.12703

Reviewer #2: Arai et al. use otolith stable isotope ratios (d13C and d18O) to assess contingent mixing and individual origin of Scomber scombrus over several years spanning multiple decades. They developed a machine-learning multi-model ensemble classifier using Bayesian model averaging and then integrate the predictions obtained with continuous isoscapes to estimate the probability of origin across two spatial domains (northern and southern contingents), identifying geographic nursery hotspots and geographic shifts over time. The manuscript is well written and clearly structured, however there are several aspects of this work that need to be addressed before this manuscript can be published.

1. Authors should provide more easily understandable information (in a table?) on the number of samples used for each step. The nominal approach used a number of individuals which varied among years while the number of samples used in the isoscapes is different.

2. Authors need to provide classification accuracies for each year of the contingent classification baseline using known-origin age-1 juveniles. Based on Fig. 3, where for some years otolith stable isotope ratios apparently lacked differences and thus discrimination was likely very low. This is a key step as the continuous approach using the isoscapes relies on these classifications and any errors or misclassifications are carried over to the next analysis.

3. The framework, in particular the isoscape part, is complicated and the advantages are rather limited. Unknown samples first need to be assigned to one of the contingents thus there needs to be a baseline (reference library), and then based on this first assignment they are assigned to a geographic location using the isoscapes. Why do authors use two isoscapes? Have they tried developing just one isoscape that covers the whole distribution range of the two contingents? This would allow to assign unknown individuals to the geographic locations and compare the results of this continuous approach based only on d18O to the results obtained using the nominal approach using both d13C and d18O and the baseline samples. Furthermore this lack of discrimination power then is replicated in the geographic assignments that are based on less than 10 individuals.

4. Authors state that the BMA approach provided the best results but based on the parameters used to assess model performance (Fig. 4), the random forest analysis performed equally to the combined BMA approach which makes the additional step of using all the different models rather superfluous. Differences between the BMA and RF approaches are minimal based on the results presented.

5. Any conclusions of spawning/nursery hotspots based on less than 10 individuals is not feasible and plotting nursery habitat use over several 100s of km2 based on 1 or 2 individuals may lead to misinterpretation and biases.

Minor comments

Abstract needs to be thoroughly revised. As is it is not evident that two isoscapes were used ans some of the number (i.e. percentage of baseline correct classification) only appear in the introduction.

Line 73: delete “into account”.

Line 80-81: rephrase since machine learning classification is also possible with a continuous reference baseline.

Line 160: More information regarding sample size per year and location is needed.

Line 341: You need to demonstrate these significant differences by presenting the classification accuracies among contingent per year.

Lines 420-425: How can you detect trends when your analysis is based on a few years within a decade and more importantly on a handful of individuals. This is more likely to be variability than a consistent trend.

Lines 510-512: Discrimination power in these samples is minimal (66% accuracy not assigned), thus it is difficult to draw any solid conclusion from these results.

Figure 2 should also include the baseline development as the first step

6. PLOS authors have the option to publish the peer review history of their article (what does this mean?). If published, this will include your full peer review and any attached files.

Reviewer #1: No

Reviewer #2: No

---

## [Author Response · Author response to Decision Letter 0]

4 Apr 2023

Point-by-point response for Arai et al (PONE-D-23-01598)

We thank the editor and two reviewers for their helpful comments and suggestions. We have made major modifications to our manuscript and have responded to all the reviewer comments below. Our responses to the comments are shown in blue.

Journal Requirements:

We have followed all PLOS ONE’s style requirements.

“KA was financially supported by a fellowship from Japan Student Services Organization.”

“Project funding was provided by Fisheries and Oceans Canada (DFO) under grant number F5211-190215 to DHS. The funders had no role in study design, data collection and analysis, decision to publish, or preparation of the manuscript.”

We have omitted the funding-related text from the manuscript. Please modify the funding statement as follows:

“Project funding was provided by Fisheries and Oceans Canada (DFO) under grant number F5211-190215 to DHS. The funders had no role in study design, data collection and analysis, decision to publish, or preparation of the manuscript. KA was financially supported by a fellowship from Japan Student Services Organization.”

We have provided all otolith oxygen and carbon stable isotope measurement data as the “minimal data set” in the Dryad repository as a private-for-peer review: https://datadryad.org/stash/share/EyXRh_nG6O-hD4wsS67TWCPRwdhLYkcEIRxiURX8cGs

We have provided all otolith oxygen and carbon stable isotope measurement data as the “minimal data set” in the Dryad repository as a private-for-peer review: https://datadryad.org/stash/share/EyXRh_nG6O-hD4wsS67TWCPRwdhLYkcEIRxiURX8cGs

5. We note that Figures 1, 5, 6, 7 and 8 in your submission contain map images which may be copyrighted. All PLOS content is published under the Creative Commons Attribution License (CC BY 4.0), which means that the manuscript, images, and Supporting Information files will be freely available online, and any third party is permitted to access, download, copy, distribute, and use these materials in any way, even commercially, with proper attribution. For these reasons, we cannot publish previously copyrighted maps or satellite images created using proprietary data, such as Google software (Google Maps, Street View, and Earth). For more information, see our copyright guidelines: http://journals.plos.org/plosone/s/licenses-and-copyright.

 a. You may seek permission from the original copyright holder of Figures 1, 5, 6, 7 and 8 to publish the content specifically under the CC BY 4.0 license. 

We note that Figures 1, 5, 6, 7 and 8 in your submission contain map images which may be copyrighted.

Figures 1, 6, 7, and 8 were generated using the R package “rworldmap” (License: GPL (>=2)), which uses map data from Natural Earth (public domain, https://www.naturalearthdata.com/about/terms-of-use/).

We further acknowledge that the GPL license in the R package “rworldmap” only pertains to the modification and redistribution of the source code of the package, which we have not done in our manuscript. The package itself uses map data from Natural Earth, a public domain, and not copyrighted. Hence, we believe the copyright holder’s consent is not required for the publication of the manuscript (for example, processing the text of the article in MS word does not imply that the article is copyrighted by Microsoft). We, however, included the following text in the captions for Figures 1, 6,7,8, and S1: “The map was created using the rworldmap package in R [64].”

We have provided all supporting information figure captions at the end of our manuscript.

 

Reviewer #1

This study proposed and described a novel framework to delineate the geographic origin and nursery hotspots over a spatiotemporal scale for the Northwest Atlantic mackerel by means of otolith stable isotope concentrations, which could be a suitable and interesting approach for other marine species as well.

This paper presents the results of original research with an important contribution to the field of stock management informed otolith microchemistry presented as a high-quality manuscript which significant importance for the broad readership of PLOS ONE journal. The manuscript contains all the relevant information on the field, and is easy to follow, the motivation behind the work is clearly described, with experiments, statistics, and other analyses are performed to a high technical standard and are described in sufficient detail, with straight and sound conclusions that are supported by the data. The manuscript is easy to follow and is presented in presented in an intelligible fashion and is written in standard English. Authors do state that meet all applicable standards for the ethics of experimentation and research integrity, and that all data are fully available without restriction.

However, there are some minor issues that could be improved in my point of view before its publication as a high impact article. I have provided general comments and suggestions for each section of the manuscript, followed by specific comments to the authors that should be quite straightforward to address, I hope authors find them useful. If authors can address these minor issues, I do believe that this manuscript will be of great value as a publication for PLOS ONE journal. Finally, my most sincere congratulations for this nice piece of work, this is good science that needs to be shared!

We thank the reviewer for their careful reading of the manuscript and suggestions for improvement. Please see below for our responses to the reviewer’s specific comments.

INTRODUCTION

The introduction provides the reader an adequate background and clearly described the motivation behind this work. However, I do fill that too much weight is given (lines 86-99) on how the analysis has been done, which is perfectly described in M&M sections, and I do miss few more background on otolith stable isotopes composition, e.g. how oxygen composition of otolith aragonite responds to variations in the environment and why it is useful. If you could balance the text between both topics it would be very nice.

R1-1: We followed the reviewer’s advice and included the following paragraph in the Introduction section on the characteristics of otolith stable isotope composition and how these tracers could be suitable for this study. 

L81–90: “Oxygen and carbon stable isotopes (�18O/�13C values) in otoliths (or ear stones) have been widely employed as a natural marker to study migration and connectivity of marine fishes (e.g., [10,29–32]). Otolith aragonite forms near isotopic equilibrium with the oxygen isotopic composition of ambient water, which generally varies globally as a function of salinity [33]. The otolith–water oxygen isotopic fractionation has also been shown to be temperature-dependent, with minor differences across species [34–38]. Otolith �13C reflects the carbon isotopic composition of dissolved inorganic carbon (DIC) and the diet source [34,39,40], with the proportion of dietary carbon in the otolith controlled by the metabolic rate in a temperature-dependent way [41]. As a result of these features, otolith �18O and �13C exhibit a predictable spatial gradient, making them suitable for the continuous (isoscape) approach in marine environments.”

We also shortened the Introduction section by removing extra information on the methodological aspects of the combined approach.

Line 53: I would suggest replacing “internal tags” for something like “natural markers” as internal tags usually refers to fish tagging devices and can be confusing.

R1-2: Modified to “natural markers” as suggested by the reviewer (L54).

Line 54: I would suggest starting the second sentence with an “Here” or sth similar to emphasize that you are still talking about chemical markers in biogenic tissues, as if not, it could be again misunderstood as the most common approach for tracking highly migratory fish is by archival or pop up tagging. Something like “Here, the most common approach consists of reconstructing…”

R1-3: Change accepted (L55).

Line 63: You can cite Jones et al. 2017 here to reinforce this statement.

Jones, C. M., Palmer, M., & Schaffler, J. J. (2017). Beyond Z ar: the use and abuse of classification statistics for otolith chemistry. Journal of Fish Biology, 90(2), 492-504.

R1-4: We thank the reviewer for their suggestion. We have cited the paper in L64.

Lines 63-66: This is true because most older papers used traditional approaches, but I would also note that there is a trend towards change e.g.

Wright, P. J., Régnier, T., Gibb, F. M., Augley, J., & Devalla, S. (2018). Assessing the role of ontogenetic movement in maintaining population structure in fish using otolith microchemistry. Ecology and Evolution, 8(16), 7907-7920.

Zhang, C., Ye, Z., Li, Z., Wan, R., Ren, Y., and Dou, S. (2016). Population structure ofJapanese Spanish mackerel Scomberomorus niphonius in the Bohai Sea, the Yellow Sea and the East China Sea: evidence from random forests based on otolith features. Fisheries Science 82, 251–256. doi:10.1007/S12562-016-0968-X

Artetxe-Arrate, I., Fraile, I., Crook, D. A., Zudaire, I., Arrizabalaga, H., Greig, A., & Murua, H. (2019). Otolith microchemistry: a useful tool for investigating stock structure of yellowfin tuna (Thunnus albacares) in the Indian Ocean. Marine and Freshwater Research, 70(12), 1708-1721.

R1-5: We thank the reviewer for their suggestion. We have cited the papers in L67.

Lines 72-73: Not sure if completely agree, don’t always explicitly incorporate all sources of variation into account. E.g. individual physiology for stable isotope incorporation

R1-6: We agree with the reviewer and replaced “all” with “multiple” (L73).

Line 80: Perhaps “Combined” instead of “Hybrid”?

R1-7: Modified to “combined” as suggested by the reviewer (L91).

Line 82: I think that is not correct to use the term “highly migratory marine species here”. Highly migratory species are legally defined as those listed in Annex 1 of UNCLOS (https://www.un.org/depts/los/convention_agreements/texts/unclos/annex1.htm) and Atlantic mackerel is not part of this list. Please adapt this sentence here and elsewhere in the manuscript.

R1-8: We agree with the reviewer and replaced the term “highly migratory” with “migratory” here (L93) and elsewhere in the manuscript.

MATERIAL AND METHODS

The section provides enough detail on the technique, equations and statistical approaches used so that other researchers can reproduce them. I just miss a little bit more information regarding the Study species, such as how it is managed (e.g. Atlantic mackerel managed separately in US and Canada?) and spawning conditions that feel important to known, such as when are they capable of spawning, age or length, so that then it is better understood why you separate age-1 as baseline and age>2 for assignment purposes. I also would recommend to include a summary Table with the individuals that have been analysed, giving information numbers in the, 201 baseline, which are for assignment, how much fish belong to each year, how much fish were collected from each fishery etc and also present the age-0 juvenile samples used for testing the accuracy of the isoscape.

R1-9: We followed the suggestion by the reviewer and included the following statement describing the management, spawning conditions, and biomass differences of the two contingents:

L116–124: “The Northwest Atlantic mackerel population consists of two sub-components: the northern and southern “contingents” that exhibit distinct spawning and nursery regions, as well as seasonal migration patterns that transverse the border between the US and Canada [49,50]. The two nations separately manage jurisdictional fisheries with Canada assessing the northern contingent as a single stock, and the US treating the two contingents as components of a single transboundary stock [47,48]. While the estimated median age-at-maturity is about 2 years for both contingents, the spawning stock biomass of the northern contingent is roughly an order of magnitude larger than that of the southern contingent [47,48,51,52].”

We also included a summary table showing sample size data in the supplementary information (S1 Table). A similar summary sample size table for age-0 juveniles was included in the supplementary information as well (S3 Table).

Line 102: See comment above about “highly migratory”, perhaps you can find another way to explain this, or go a bit further and explain here the migrations that Atlantic mackerel undergoes (e.g. from spawning grounds in XX towards feeding grounds in XX).

R1-10: We agree with the reviewer and replaced the term “highly migratory” with “migratory” here (L111) and elsewhere in the manuscript. We also described in the “study species” section in the Methods (L128–131) and in Figure 1 that the mixing region extends as far south as the Mid-Atlantic Bight in US shelf waters, illustrating the extensive seasonal migration patterns of the Northwest Atlantic mackerel, particularly for the northern contingent.

Line 109: What does “dominant” imply here? Higher production, biomass…?

R1-11: We have included the following sentence:

L121–124: “While the estimated median age-at-maturity is about 2 years for both contingents, the spawning stock biomass of the northern contingent is roughly an order of magnitude larger than that of the southern contingent [47,48,51,52].”

Line 118: I would suggest removing “(or ear stone)” from here, and if you prefer, to move this to the Introduction.

R1-12: Change accepted (L132).

Lines 137 and 139: I cannot see what you are referring to in Fig.1. Perhaps a table could help.

R1-13: We followed the reviewer’s suggestion and included a supplementary table showing a detailed sample size of Northwest Atlantic mackerel otoliths (S1 Table).

Line 142: Why do you say that age-1 are known origin? Please can you add more information that support this statement, e.g. they do not migrate until age X, or they have shown to have limited movements outside nursery areas….

R1-14: We agree with the reviewer that this assumption needs extra justification. We have included the following statement describing the localized migration patterns of juvenile mackerel from size distribution and tagging studies (Sette 1950; Uriarte et al. 2001).

L159–163: “We assumed that the collection sites of age-1 juvenile samples represented their natal habitats and that the exchange of individuals between the two contingents before reaching adulthood was limited. These assumptions are supported by evidence from size distribution analysis and extensive tagging programs that suggest localized migration patterns of juvenile mackerel [50,65].”

Lines 158-159: Can you explain more on the implications of including some age-2 northern contingents in the baseline? Were they used again for assignments or not? Does Atlantic mackerel show some spawning site fidelity?

R1-15: As age-1 juveniles of the northern contingent are rarely encountered by Canadian commercial fisheries, age-2 samples collected in the summer Canadian fishery in the Gulf of St. Lawrence were used to complement year-classes for which age-1 samples of the northern contingent were insufficient in sample size. Age-2 northern contingent samples were only used to establish the baseline and were not used again for natal assignment analysis. Further, we believe that the inclusion of age-2 fish as a baseline for the northern contingent is supported by evidence that immigration of the southern contingent into Canadian waters is limited (Sette 1950, Moores 1975). We have included the following statement:

L179–181: “The inclusion of age-2 fish in the northern contingent baseline is supported by evidence that immigration of the southern contingent into Canadian waters is uncommon [50,60].”

Lines 161: Was Multivariate normality, linearity and multicolinearity and homoscedasticity of otolith �18O and �13C data tested prior MANOVA analyses?

R1-16: All assumptions of MANOVA were verified based on scatterplots of �18O and �13C on each grouping variable (i.e., contingent, year-class) prior to the analysis. We further assessed the statistical significance of MANOVA using Pillai’s trace which has been shown to be most robust to violations of assumptions (Scheiner 2001, Finch 2005). We have modified the statement describing MANOVA in the Methods accordingly (L184–187).

To further verify the results of MANOVA, we conducted a parallel analysis using two univariate ANOVAs. The ANOVAs were applied to models fitted using generalized least squares (using the “varIdent” function in the “nlme” package), which allowed for different variances per treatment (i.e., heteroscedastic). The p-values were adjusted for multiple comparisons using Benjamini and Hochberg correction. Results from univariate ANOVAs matched those from MANOVA (Table A). Note the result of the additional univariate test was not included in the main text.

Table A. Summary table comparing MANOVA and univariate ANOVAs on �18O and �13C allowing unequal variances per treatment.

Test Contingent Year-class Interaction

MANOVA ***p < 0.001 ***p < 0.001 ***p < 0.001

Univariate ANOVA d18O ***p < 0.001 ***p < 0.001 ***p < 0.001

Univariate ANOVA d13C ***p < 0.001 ***p < 0.001 ***p < 0.001

Line 245: If possible, could you please add which depth range are you considering here as “subsurface” (e.g 0-5 m depth)..

R1-17: Modified to “Data at 5 m depth were used…(L275)”

RESULTS

Line 341-343: Are there significant differences between contingents in every year? I think it will be nice and interesting to analyze this option too, as there are years (e.g. 1973, 1974, 2011, 2016…) were there it seems not to be differentiation, and this is also important for the discussion.

R1-18: We have followed the suggestion by the reviewer and conducted a post-hoc multivariate pairwise comparison between contingents nested within each year-class using the “mvpaircomp” function in the “biotools” package. A Bonferroni adjustment for multiple comparisons was used. �18O and �13C were significantly different between contingents for all year-classes except for 1974 and 2016. These results were included in Figure 3 along with cross-validated accuracy to show otolith stable isotope distinction for each year-class. We have updated the Methods (L188–190) and Results (L375–376) sections, and Figure captions (L385–387) accordingly.

We have also included the following statement in the Discussion section regarding year-classes with weak isotopic distinction:

L548–553: “It is important to exercise caution when interpreting contingent mixing levels, as some year-classes (e.g., 1974, 2016) exhibited weak stable isotopic distinction (Fig. 3). Nonetheless, the aggregated year-class baseline compensates for temporal variation and enables year-classes with less distinct isotopic signatures to draw information from those that are more distinct, bringing them closer to the global mean. Furthermore, a conservative 0.7 classification threshold was applied to reduce misclassification of unknown adult samples.”

Line 343-344: Less distinction or no distinction?

R1-19: A post-hoc multivariate pairwise comparison between contingents nested within each year-class showed that year-classes 1973 and 1974 were “less” distinct compared to those from year-classes 1998–2000 (L375–380, Figure 3). Please see the response “R1-18” for more details on multivariate pairwise comparisons.

Line 347: From the Figure 3 you cannot appreciate the fact that northern contingents from 1973 1974, 2011,2013 and 2014 for instance had lower oxygen values.

R1-20: We agree with the reviewer that the lower oxygen values are not immediately apparent in some year-classes. We included the word “generally” in the sentence (L379) to emphasize the general trend in otolith stable isotope composition in year-classes 1973, 1974, and 2011–2016.

Lines 357: What is the threshold for low moderate and high? Not sure if needed but think it will improve readers experience.

R1-21: We appreciate the reviewer for their suggestion. We grouped individual classifiers into three broad categories based on cross-validated mean classification accuracy as follows: low (accuracy < 0.6: LDA, LR, and NB), moderate (accuracy < 0.76: QDA and DT), and high (accuracy > 0.76: MELR, SVM, ANN, RF, and kNN) performance (Fig 4). Note, however, that classification performance was generally consistent across all three metrics. The Results section was modified accordingly (L391–395).

Line 395-396: From figure 5 it seems that northern contingents were not prevalent in 2011, 2012 and 2016, please clarify this.

R1-22: We agree with the reviewer and modified the sentence as follows:

L433–434: “Northern contingent mixing within US shelf waters was prevalent in most year-classes, except for year-classes 2011, 2012, and 2016, yet mixing levels varied greatly over the past four decades.”

Lines 400-403: This paragraph is a little bit confusing for me, because I do not have very clear were this numbers come from.

R1-23: To better clarify this paragraph, we have modified Table 1 to include the percentage of unassigned adult samples for the BMA classifier as well as from previous methods.

Figure 6: I think it could facilitate to the reader if you can add the locations you mentioned in the text in lines 415-425. This is just a suggestion, but as you have 4 x3 plots and a blank space, you can use this black space to draw the same map without predictions but with location names, so then it is easier to follow the rest of the maps. If not, one can always return to Figure 1.

R1-24: We followed the suggestion by the reviewer and added the locations in Figures 6 and 8. Figure captions were modified accordingly.

DISCUSSION

Authors place into context observed results and highlight the most important and discuss the implications of this study into a broader context, which is very interesting and nice. It provides a good overview and closure of the manuscript, with conclusions aligning with the aim of the proposed research. However, I do miss some more further discussion on the potential limitations of the approach and some mention to the Suess effect into otolith C13 data.

R1-25: We followed the reviewer’s advice and included the following paragraph in the Discussion section on the potential limitations of the combined approach:

L612–624: “The combined approach proposed here requires a reference baseline and a species-specific otolith �18O isoscape generated from temperature and salinity data through a species-specific fractionation equation. In particular, �18O of ambient seawater (�18Oseawater) was determined from salinity data which could be problematic for species that inhabit regions that are heavily influenced by multiple sources of freshwater inputs, in which the salinity–�18Oseawater relationship may be highly dynamic [97]. In this case, direct �18Oseawater measurements should be preferred for otolith �18O isoscape development, although the spatiotemporal resolution could be reduced [98]. Furthermore, because otolith �13C is influenced by an unknown combination of ambient DIC, food source, and metabolic rates [34,39,40], we were unable to establish a spatiotemporally accurate otolith �13C isoscape. Instead, we employed otolith �13C data as a predictor in the nominal approach, which was then used to inform the continuous approach. As global-scale estimations of these values have become more readily available [99–103], a multi-isotope isoscape model could be developed and applied to provide a more accurate geographic origin estimate in marine species [24,26].”

We further included the following statement in the Discussion section regarding the Suess effect:

L604–609: “There was no clear inter-decadal pattern in the measured juvenile baseline for otolith �13C values (Fig. 3); however, the decline in atmospheric �13C values has been linked to a significant declining trend in baseline otolith �13C in Atlantic bluefin tuna (i.e., Suess effect) [81]. These strong temporal patterns in otolith stable isotopes further highlight the importance of accounting for such variation in the baseline for accurate natal assignment.”

Line 495: I suggest replacing “hybrid” by “combined”

R1-26: Change accepted (L537).

Line 529: Perhaps “primarily driven by changes in temperature”

R1-27: Change accepted (L577).

Line 559: I suggest replacing “tissue” here by “biogenic”, “hard-calcareous structures” or others here.

R1-28: Replaced “tissue” with “biogenic”, as suggested (L625).

Line 562: But see Martino et al. 2022

Martino JC, Trueman CN, Mazumder D, Crawford J, Doubleday ZA. The universal imprint

of oxygen isotopes can track the origins of seafood. Fish Fish. 2022. doi:10.1111/faf.12703

R1-29: Change accepted (L628).

 

Reviewer #2

Arai et al. use otolith stable isotope ratios (d13C and d18O) to assess contingent mixing and individual origin of Scomber scombrus over several years spanning multiple decades. They developed a machine-learning multi-model ensemble classifier using Bayesian model averaging and then integrate the predictions obtained with continuous isoscapes to estimate the probability of origin across two spatial domains (northern and southern contingents), identifying geographic nursery hotspots and geographic shifts over time. The manuscript is well written and clearly structured, however there are several aspects of this work that need to be addressed before this manuscript can be published.

Authors should provide more easily understandable information (in a table?) on the number of samples used for each step. The nominal approach used a number of individuals which varied among years while the number of samples used in the isoscapes is different.

R2-1: We followed the reviewer’s advice and provided the sample size used at each step of the analysis in Figure 2. We further included a supplementary table showing detailed sample size data (S1 Table).

Authors need to provide classification accuracies for each year of the contingent classification baseline using known-origin age-1 juveniles. Based on Fig. 3, where for some years otolith stable isotope ratios apparently lacked differences and thus discrimination was likely very low. This is a key step as the continuous approach using the isoscapes relies on these classifications and any errors or misclassifications are carried over to the next analysis.

R2-2: We followed the suggestion by the reviewer and provided cross-validated classification accuracies for each year-class in Figure 3. We also conducted a post-hoc multivariate pairwise comparison between contingents nested within each year-class, which showed that �18O and �13C were significantly different between contingents for all year-classes except for 1974 and 2016. These results were included in Figure 3 along with cross-validated accuracies. We have updated the Methods (L188–190) and Results (L375–376) sections, and Figure captions (L385–387) accordingly.

Regarding misclassification carry-overs to isoscape assignment, we only conducted geographic assignment to samples that were above the 0.7 probability threshold (please see Figure 2). We employed a conservative 0.7 threshold to account for the incomplete separation between the baseline stable isotope composition of the two contingents in some year-classes which helped prevent misclassification of unknown adult samples but also reduced any errors associated with geographic natal assignments.

The framework, in particular the isoscape part, is complicated and the advantages are rather limited. Unknown samples first need to be assigned to one of the contingents thus there needs to be a baseline (reference library), and then based on this first assignment they are assigned to a geographic location using the isoscapes. Why do authors use two isoscapes? Have they tried developing just one isoscape that covers the whole distribution range of the two contingents? This would allow to assign unknown individuals to the geographic locations and compare the results of this continuous approach based only on d18O to the results obtained using the nominal approach using both d13C and d18O and the baseline samples. Furthermore this lack of discrimination power then is replicated in the geographic assignments that are based on less than 10 individuals.

R2-3: Two contingent-specific isoscapes were necessary to account for temporal differences in spawning dates and the first growing season between the two contingents. This required us to average temperature and �18Oseawater for three consecutive months from the known peak spawning date for each contingent: May–July for the southern contingent, and June–August for the northern contingent. We believe that a single isoscape developed over the entire growing season (e.g., April–October) of the Northwest Atlantic mackerel will not adequately represent the oxygen isotope values incorporated in otolith material within the first annulus of each contingent. A detailed explanation of developing contingent-specific isoscapes is described in L250–255 and L263–267 in the Methods section.

While we acknowledge the importance of comparing results from two independent approaches (nominal and continuous), the combined approach (i) included the informative �13C predictor in the natal assignment process which is challenging to model across space; and (ii) allowed fine-scale natal assignments on predefined boundaries which could be particularly informative for management and conservation applications. We have further included a paragraph in the Discussion section acknowledging the potential limitations of this approach (L612–624).

Authors state that the BMA approach provided the best results but based on the parameters used to assess model performance (Fig. 4), the random forest analysis performed equally to the combined BMA approach which makes the additional step of using all the different models rather superfluous. Differences between the BMA and RF approaches are minimal based on the results presented.

R2-4: We acknowledge that the difference between the BMA and RF models was small. However, the BMA approach outperformed the RF model in both the AUC and log-loss metrics. Further, to the best of our knowledge, the BMA approach has never been applied to otolith-chemistry-based stock discrimination studies. Previous studies often select the “best” model based on a metric (e.g., accuracy or AUC), which ignores any uncertainty associated with the model selection process and introduces the model selection bias (Lukacs et al. 2010; https://doi.org/10.1007/s10463-009-0234-4). As modern machine learning algorithms become more accessible, we believe it is important to present this framework and its applicability for future otolith-based stock discrimination studies.

Any conclusions of spawning/nursery hotspots based on less than 10 individuals is not feasible and plotting nursery habitat use over several 100s of km2 based on 1 or 2 individuals may lead to misinterpretation and biases.

R2-5: We agree with the reviewer that plotting hotspots from small samples could lead to biased findings. However, one of the key objectives of this study was to demonstrate the feasibility of the new integrated framework, rather than to pinpoint the nursery hotspot for this species, which is described in the Introduction section (L91–107). We believe it is important that all the information from the new approach should be preserved and displayed. Thus, instead of removing data for year-classes with small sample sizes, we kept the information but included the following statement in the Discussion section:

L581–583: “Furthermore, nursery hotspot predictions from year-classes with small sample sizes (e.g., 1973, 2000, 2016) are prone to bias and may not be indicative of the contingent-level nursery habitat and should be considered with caution.”

Furthermore, we have clearly indicated the number of samples used for each year-class in Figure 8 to prevent any misinterpretation and biases.

Abstract needs to be thoroughly revised. As is it is not evident that two isoscapes were used and some of the numbers (i.e. percentage of baseline correct classification) only appear in the introduction.

R2-6: We followed the reviewer’s suggestion and revised the Abstract to show that two isoscapes were used:

L40–41: “Nominal predictions were integrated into mackerel-specific otolith oxygen isoscapes developed independently for Canadian and US waters.”

We have also indicated in the Abstract the cross-validated mean classification accuracy across all year-classes (L36–38).

Line 73: delete “into account”.

R2-7: Change accepted (L74).

Line 80-81: rephrase since machine learning classification is also possible with a continuous reference baseline.

R2-8: We followed the reviewer's suggestion and rephrased the statement as follows:

L91–94: “Here, we present a combined approach that integrates predictions from a nominal assignment approach (machine learning classification) with a continuous assignment approach (isoscapes) to uncover the connectivity and spatial structure of a migratory species in the Northwest Atlantic Ocean over multi-decadal time scales.”

Line 160: More information regarding sample size per year and location is needed.

R2-9: We followed the reviewer’s suggestion and included a supplementary table showing detailed sample size data (S1 Table).

Line 341: You need to demonstrate these significant differences by presenting the classification accuracies among contingent per year.

R2-10: We followed the suggestion by the reviewer and provided cross-validated classification accuracies for each year-class in Figure 3. We also conducted a post-hoc multivariate pairwise comparison between contingents nested within each year-class, which showed that �18O and �13C were significantly different between contingents for all year-classes except for 1974 and 2016. These results were included in Figure 3 along with cross-validated accuracies. We have updated the Methods (L188–190) and Results (L375–380) sections, and Figure captions (L382–388) accordingly.

Lines 420-425: How can you detect trends when your analysis is based on a few years within a decade and more importantly on a handful of individuals. This is more likely to be variability than a consistent trend.

R2-11: We agree with the reviewer and replaced the word “decadal trend” with “temporal variation”. We have modified the statement in the Results section as follows:

L458–461: While the broad spatial pattern in �18Ooto values across the Northwest Atlantic Ocean persisted over four decades, strong temporal variation was observed at regional scales, particularly in the southern GSL, the western GOM, SNE, and MAB regions.

Lines 510-512: Discrimination power in these samples is minimal (66% accuracy not assigned), thus it is difficult to draw any solid conclusion from these results.

R2-12: We agree with the reviewer that the classification accuracy of the 1973 and 1974 year-classes was relatively low, and contingent mixing level estimates could be less accurate compared to other year-classes. Thus, we replaced the word “dominance” with “presence” (L558) and included a statement in the discussion acknowledging the small sample size and high unassignment rates for this period (L559–562). Still, the conservative 0.7 classification threshold excluded individuals of low contingent membership probability, and only those that showed a clear northern contingent signature were included in the analysis. We believe that these results support the claim of prevalent northern contingent mixing and possible subsidy to the large historical fishery.

We further included the following statement in the Discussion section regarding contingent mixing level estimates from year-classes with weak isotopic separation:

L548–553: “It is important to exercise caution when interpreting contingent mixing levels, as some year-classes (e.g., 1974, 2016) exhibited weak stable isotopic distinction (Fig. 3). Nonetheless, the aggregated year-class baseline compensates for temporal variation and enables year-classes with less distinct isotopic signatures to draw information from those that are more distinct, bringing them closer to the global mean. Furthermore, a conservative 0.7 classification threshold was applied to reduce misclassification of unknown adult samples.”

Figure 2 should also include the baseline development as the first step.

R2-13: We followed the suggestion by the reviewer and included the baseline development as “step A” in Figure 2. We further included the isoscape validation process as “step G”. Figure captions have been modified accordingly (L206–221).

---

## [Decision Letter · Decision Letter 1]

2 May 2023

Integrating machine learning with otolith isoscapes: reconstructing connectivity of a marine fish over four decades

PONE-D-23-01598R1

Dear Dr. Arai,

We’re pleased to inform you that your manuscript has been judged scientifically suitable for publication and will be formally accepted for publication once it meets all outstanding technical requirements.

Kind regards,

Antonio Medina Guerrero, Ph.D.

Academic Editor

PLOS ONE

Additional Editor Comments (optional):

Reviewers' comments:

Reviewer's Responses to Questions

**Comments to the Author**

1. If the authors have adequately addressed your comments raised in a previous round of review and you feel that this manuscript is now acceptable for publication, you may indicate that here to bypass the “Comments to the Author” section, enter your conflict of interest statement in the “Confidential to Editor” section, and submit your "Accept" recommendation.

Reviewer #1: All comments have been addressed

Reviewer #2: All comments have been addressed

2. Is the manuscript technically sound, and do the data support the conclusions?

Reviewer #1: Yes

Reviewer #2: Yes

3. Has the statistical analysis been performed appropriately and rigorously? 

Reviewer #1: Yes

Reviewer #2: Yes

4. Have the authors made all data underlying the findings in their manuscript fully available?

Reviewer #1: Yes

Reviewer #2: Yes

5. Is the manuscript presented in an intelligible fashion and written in standard English?

Reviewer #1: Yes

Reviewer #2: Yes

6. Review Comments to the Author

Reviewer #1: The authors have done a great job in responding to ALL the minor comments suggested and adapting them in the text. Congratulations to the authors, it has been a pleasure to learn from their manuscript.

Reviewer #2: (No Response)

7. PLOS authors have the option to publish the peer review history of their article (what does this mean?). If published, this will include your full peer review and any attached files.

Reviewer #1: No

Reviewer #2: No

---

## [Editor Report · Acceptance letter]

19 May 2023

PONE-D-23-01598R1 

Integrating machine learning with otolith isoscapes: reconstructing connectivity of a marine fish over four decades 

Dear Dr. Arai:

I'm pleased to inform you that your manuscript has been deemed suitable for publication in PLOS ONE. Congratulations! Your manuscript is now with our production department. 

Kind regards, 

on behalf of

Dr. Antonio Medina Guerrero 

Academic Editor

PLOS ONE